# Insects evolved a monomeric histone-fold domain in the CENP-T protein family

Sundar Ram Sankaranarayanan [1], Jonathan Ulmer[1], Anna Mørch[2,3], Ahmad Ali-Ahmad [2],
Nikolina Sekulić [2,3,4] & Ines Anna Drinnenberg [1✉]

## Abstract

**The histone-fold domain (HFD) is a conserved protein interaction module that requires stabilization through a handshake interaction with an HFD partner. All HFD proteins known to date form obligate dimers to shield the extensive hydrophobic residues along the HFD. Here, we find that the lepidopteran kinetochore protein CENP-T is soluble as a monomer. We attribute this stability to a structural rearrangement, which leads to the repositioning of the HFD helix α3. This brings a conserved two-helical extension closer to the histone fold, where it takes over the position and function of the CENP-T partner CENP-W. This change has no effect on the DNA-binding ability of the lepidopteran CENP-T. Our analysis suggests that the monomeric HFD originated in the last common ancestor of insects, with a possible second independent origin in Acariformes, both of which lack CENP-W. Our study highlights an unexpected structural variation in a protein module as conserved and optimized as the HFD, providing a unique perspective on the evolution of protein structure and the forces driving it.**

**Keywords** Kinetochore; CCAN; Mitosis; Lepidoptera
**Subject Categories** Chromatin, Transcription & Genomics; Evolution & Ecology; Structural Biology

## Introduction

The histone-fold domain (HFD), a defining feature of histone proteins, is one of the most abundant and conserved DNA binding domains detected across eukaryotes and archaea and it is present in certain bacterial proteins (Talbert et al, 2019; Makarova et al, 2005; Postberg et al, 2010; Hocher et al, 2023; Henneman et al, 2018). The HFD is characterized by the presence of a long central helix flanked by two shorter helices that are connected by loops (Arents et al, 1991). The positively charged sidechains on the surface of the folded HFD favor interactions with the DNA backbone (Arents and Moudrianakis, 1995). The presence of exposed hydrophobic

residues along the helices of the HFD renders HFD proteins unstable as monomers (Karantza et al, 1995, 1996; Banks and Gloss, 2004). A handshake-like interaction with a second HFD partner is required to shield these residues and stabilize the protein. Thus, the HFD can also be considered a protein-dimerization motif in addition to its role in DNA binding (Arents et al, 1991; Arents and Moudrianakis, 1995). Besides histones, the HFD is also commonly detected in proteins involved in transcription, DNA replication and repair, chromatin remodeling, as well as at the kinetochore (Fig. 1A), wherein they also retain the dimeric state apart from their role in DNA binding (Hartlepp et al, 2005; Kamada et al, 2001; Gangloff et al, 2001; He et al, 2017).

The kinetochore is a multiprotein complex assembled on the centromere of each chromosome that connects chromosomes to spindle microtubules, enabling accurate chromosome segregation during cell division. The kinetochore has two primary layers: an inner kinetochore called Constitutively Centromere Associated Network (CCAN), which is proximal to centromeric DNA, and an outer kinetochore that attaches to the spindle microtubules (Cheeseman, 2014; Ariyoshi and Fukagawa, 2023). Two components of the CCAN, CENP-C and CENP-T, are critical to connect the inner to the outer kinetochore. CENP-C directly binds to nucleosomes containing a specialized histone H3 variant CENP-A (Falk et al, 2015; Kato et al, 2013; Carroll et al, 2010) that epigenetically marks centromeric chromatin and initiates kinetochore assembly in various eukaryotes. Via its N-terminus CENP-C, in turn, interacts with components of the outer kinetochore network (Milks et al, 2009; Przewloka et al, 2011; Screpanti et al, 2011). CENP-T binds DNA directly via a C-terminal HFD in the context of a CENP-TWSX nucleosome-like complex, and supercoils linker DNA, thereby sharing structural and functional properties of histones (Nishino et al, 2012; Takeuchi et al, 2014; Hori et al, 2008). At the CCAN, this complex partially wraps DNA and bends it through the interactions made by the positively charged residues in and upstream of the HFD-α1 (Yatskevich et al, 2022; Takeuchi et al, 2014). In addition, CENP-T also contains N-terminal motifs that bind the outer kinetochore proteins Ndc80 and Mis12 (Malvezzi et al, 2013; Huis In 't Veld et al, 2016; Nishino et al, 2012; Hori et al, 2008). Tethering CENP-T to ectopic sites on the chromosome was shown to be sufficient to recruit outer kinetochore proteins and segregate minichromosomes (Hori et al,

[1]Institut Curie, PSL Research University, Sorbonne Université, CNRS, UMR3664 Nuclear Dynamics Unit, Paris 75005, France. [2]Centre for Molecular Medicine Norway (NCMM), Nordic EMBL Partnership, Faculty of Medicine, University of Oslo, Oslo 0318, Norway. [3]Department of Chemistry, University of Oslo, Oslo 0315, Norway. [4]Department of Molecular Medicine, Institute of Basic Medical Sciences, Faculty of Medicine, University of Oslo, Oslo 0372, Norway. ✉E-mail: ines.drinnenberg@curie.fr

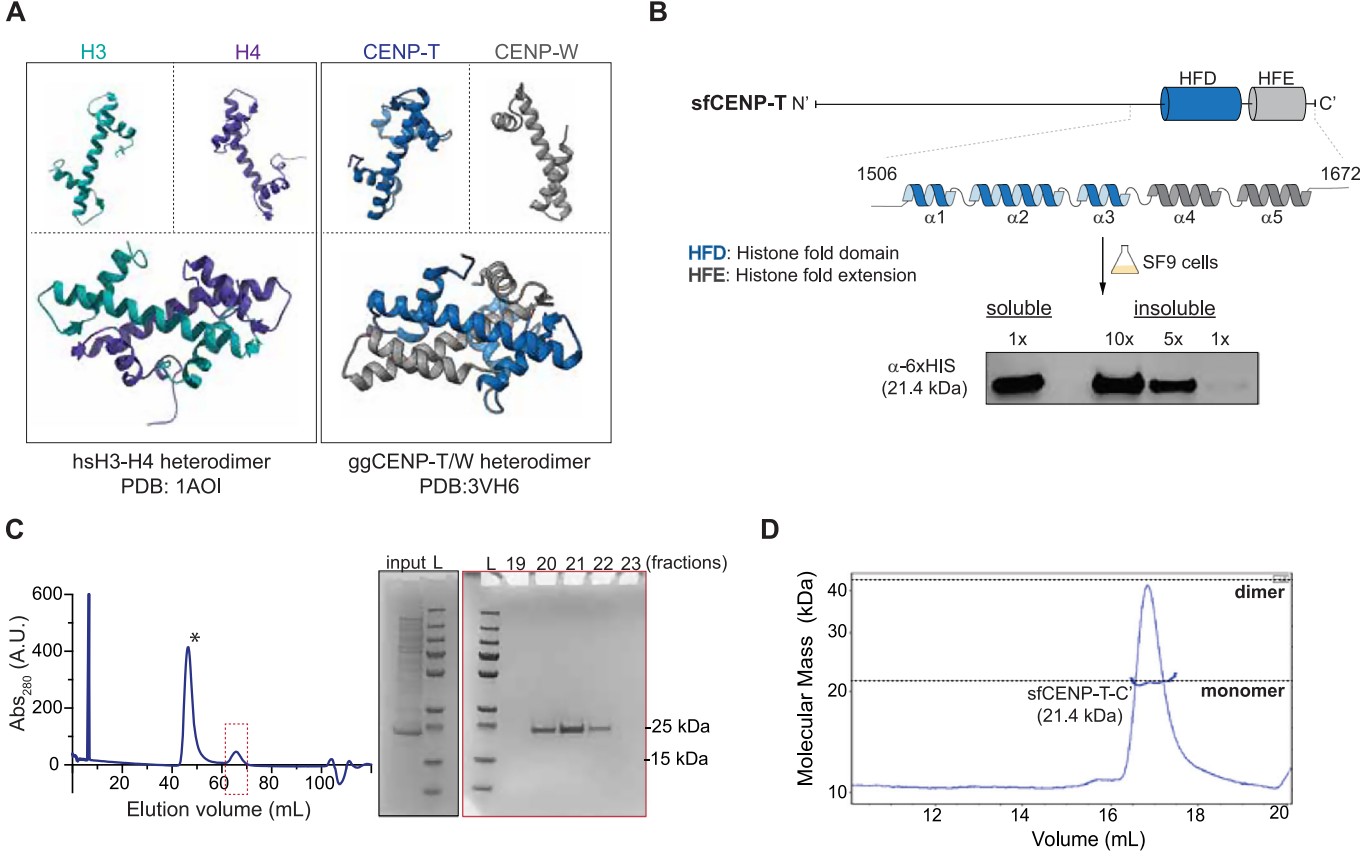

**Figure 1. Lepidopteran CENP-T is a soluble monomeric protein.**

(A) The organization of helices of a canonical HFD monomer and their physiological heterodimeric state illustrated from the known structures of histones H3-H4 from humans (PDB: 1AO1) and CENP-TW from chicken (PDB:3vh6). (B) (Top) Line diagram of sfCENP-T with the relative positions of the histone-fold domain (HFD) and the histone fold extension (HFE) marked by blue and gray cylinders, respectively. The linear structure of each of these domains is expanded. (Bottom) Analysis of soluble and insoluble fractions of SF9 cells expressing the C-terminal fragment of 6xHis-sfCENP-T[1147-1314] by western blot analysis. (C) The elute from the affinity purification step for 6xHis-sfCENP-T[1147-1314] fragment was further purified to homogeneity by size-exclusion chromatography (SEC). The plot depicts the SEC elution profile with the x and y axes indicating elution volume and absorbance at 280 nm, respectively. The peak fractions marked by an asterisk and a dotted red box were visualized by SDS-PAGE stained with Coomassie Blue (also see Fig. EV1A). The analysis of the peak marked with the asterisk is presented in the expanded version in Fig. EV1A. Input: a fraction of the elute from the affinity purification step for 6xHis-sfCENP-T[1147-1314] fragment loaded into the SEC column. L molecular weight marker. (D) Multi-angle Light Scattering coupled with SEC (SEC-MALS) of the 6xHis-sfCENP-T[1147-1314] fragment reveals the monomeric state of the protein (21.2 kDa). The molecular weight inferred from MALS is represented as a blue line overlaid on the SEC peak for this protein. Source data are available online for this figure.

2013; Schleiffer et al, 2012). Like other HFD proteins, the stability, localization, and function of CENP-T is dependent on dimerization with a HFD partner, CENP-W (Schleiffer et al, 2012; Hori et al, 2008). The strong interdependence of CENP-T and CENP-W is also reflected in their simultaneous presence or absence across the tree of life (Tromer et al, 2019). The CENP-TW heterodimer oligomerizes with the CENP-SX heterodimer to form the nucleosome-like structure that preferably binds to a ~100 bp linker DNA in the presence of nucleosomes (Nishino et al, 2012; Takeuchi et al, 2014). It should be noted that tetramerization is not essential for CENP-T function at the centromere, as CENP-T localization was not perturbed by CENP-S depletion (Amano et al, 2009; Nishino et al, 2012). However, mutations affecting the interactions between CENP-TW and DNA resulted in the loss of its kinetochore localization and defective mitosis (Nishino et al, 2012). This makes the HFD and its interactions with DNA a critical node in vertebrate kinetochore assembly. Besides DNA binding, CENP-T also

interacts with the CENP-HIKM complex, additional CCAN components that contribute to the inner kinetochore assembly (Basilico et al, 2014; McKinley et al, 2015; Pekgöz Altunkaya et al, 2016). The interface between these two complexes was resolved to be a three-helix bundle formed by the two conserved helices of the CENP-T- Histone Fold Extension (HFE or two-helical extension) and an N-terminal helix from CENP-K (Pekgöz Altunkaya et al, 2016; Hinshaw and Harrison, 2020; Zhang et al, 2020). The interaction between CENP-T and the CENP-HIKM complex is crucial for kinetochore stability and function (Pekgöz Altunkaya et al, 2016; Basilico et al, 2014; McKinley et al, 2015).

In previous studies, we conducted proteomic analyses (IP/MS) combined with remote homology predictions to get insights into the composition and assembly of kinetochores in Lepidoptera (Cortes-Silva et al, 2020). Lepidoptera are interesting because they have lost the otherwise conserved centromere marker protein CENP-A and outer kinetochore linker protein CENP-C

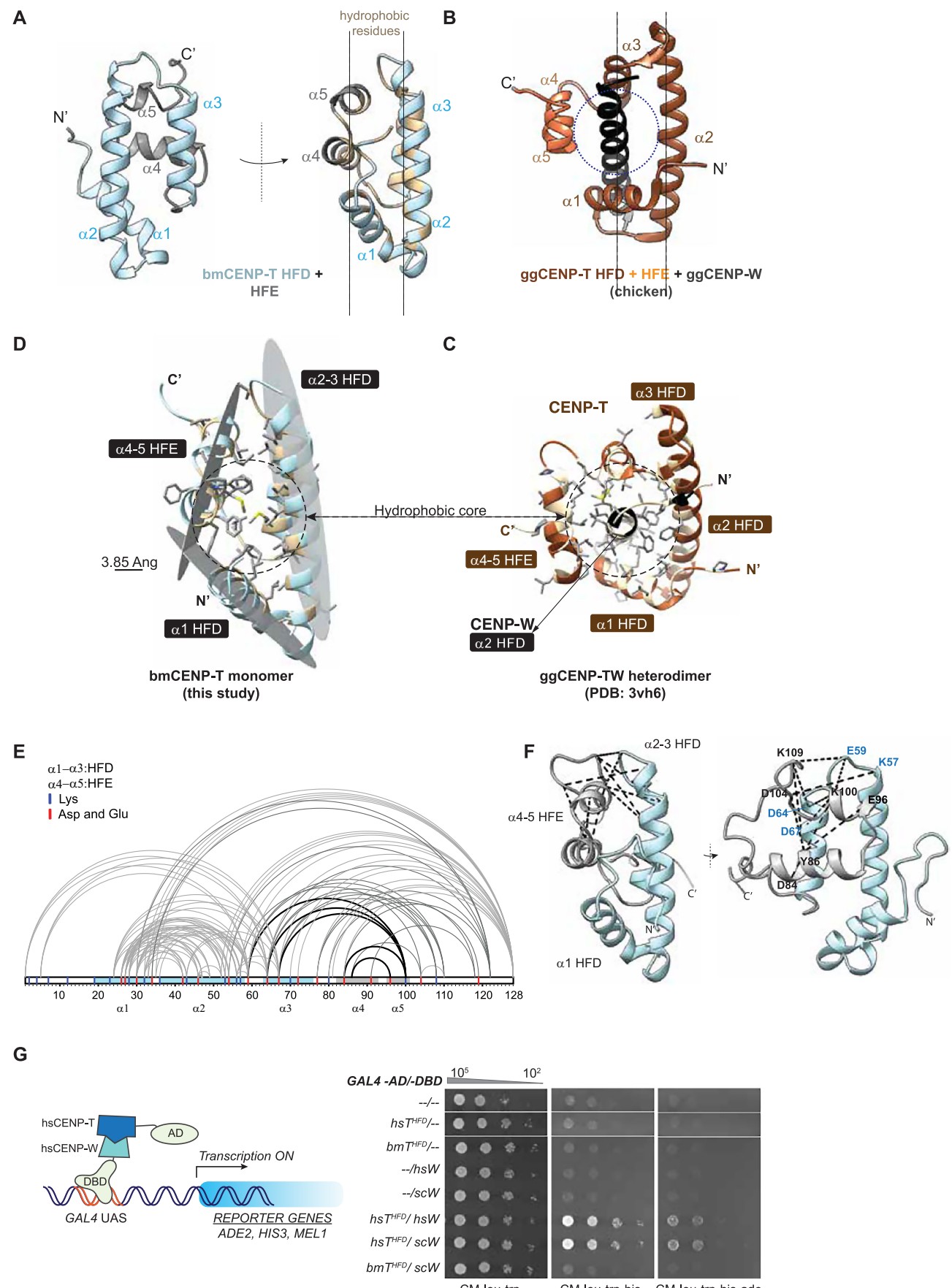

**A** bmCENP-T HFD + HFE

hydrophobic residues

**B** ggCENP-T HFD + HFE + ggCENP-W (chicken)

**D** bmCENP-T monomer (this study)

3.85 Ang

Hydrophobic core

**C** ggCENP-TW heterodimer (PDB: 3vh6)

CENP-T
α3 HFD
α2 HFD
α1 HFD
α4-5 HFE
CENP-W
α2 HFD

**E**
α1−α3:HFD
α4−α5:HFE
Lys
Asp and Glu

10  20  30  40  50  60  70  80  90  100  110  120  128
α1    α2      α3    α4  α5

**F**
α2-3 HFD
α4-5 HFE
α1 HFD
K109  E59  K57
D104  K100  E96
D64
D67
Y86
D84

**G**
hsCENP-T
hsCENP-W
AD
DBD
Transcription ON
GAL4 UAS
REPORTER GENES
ADE2, HIS3, MEL1

GAL4 -AD/-DBD
10^5    10^2
--/--
hsT^HFD/--
bmT^HFD/--
--/hsW
--/scW
hsT^HFD/ hsW
hsT^HFD/ scW
bmT^HFD/ scW

CM-leu-trp    CM-leu-trp-his    CM-leu-trp-his-ade

Figure 2.  The monomeric histone fold and the repositioned extension occlude CENP-W and stabilize CENP-T as a monomer.

(A) Front and side views of the bmCENP-T C-terminus containing the HFD and HFE as modeled by AlphaFold. Hydrophobic residues within this fragment are colored beige in the side view. The sequences upstream of the HFD in bmCENP-T were hidden in the model for clarity. (B) A comparative side view of ggCENP-TW heterodimer, highlighting the position of ggCENP-T-HFE relative to ggCENP-W and the ggCENP-T-HFD (dotted circle). For clarity, only the second helix of the ggCENP-W histone fold is shown in black. The images were adapted from PDB entry 3VH6 (Nishino et al, 2012). (C) Illustration of the proposed stabilization of bmCENP-T histone fold by hydrophobic residues present in the HFE. Three hydrophobic surfaces from α1, α2-3, and α4-5 are proposed to form a hydrophobic core. The hydrophobic residues and their side chains are colored as in (A). Scale bar, 3.85 Ang. (D) A tilted view of ggCENP-TW complex shown in (B) depicting the proximity of the side chains of hydrophobic residues from the helices of ggCENP-T and ggCENP-W. The hydrophobic residues are shaded beige in the backbone, and their side chains are represented as sticks in gray. (E, F) EDC-NHS crosslinking mass spectrometry of bmCENP-T fragment represented in a 1D plot (E) and in the 3D model (F). Crosslinks validating the proximity between HFE and HFD are highlighted as black lines, and the corresponding residues are labeled. See also Dataset EV1. (G) Schematic representation of a yeast two-hybrid assay (left panel). The transcription of reporter genes is driven by the interaction of proteins that are fused to the GAL4-AD and GAL4-DBD proteins, respectively. The known interacting proteins hsCENP-T and hsCENP-W are shown as representative examples in this schematic. Results from spot dilution assay performed with the reporter strain transformed with plasmids encoding the indicated GAL4-AD and -DBD fusions are shown in the right panel. Growth on CM-leu-trp serves as a control. Growth on CM-leu-trp-his and CM-leu-trp-his-ade dropout plates indicates positive interaction between the proteins cloned in fusion with GAL4-AD and GAL4-DBD. The plates were photographed after incubation at 30 °C for 48 h. Source data are available online for this figure.

(Drinnenberg et al, 2014). These analyses uncovered the presence of orthologs of several kinetochore components common in other eukaryotes, including a divergent homolog of CENP-T. Using *Bombyx mori* as an experimental model system, we found that much like the human CENP-T homolog, *B. mori* CENP-T (bmCENP-T) is essential for viability. Depleting bmCENP-T disrupted the localization of other kinetochore subunits, resulting in mitotic defects. Tethering bmCENP-T to ectopic sites recruited outer kinetochore proteins, such as Ndc80 and Mis12, as observed in yeast and chicken. Despite the functional conservation, we were unable to detect a potential homolog of the stabilizing partner CENP-W in Lepidoptera by homology searches or by affinity purifications of CENP-T or other conserved CCAN subunits (Cortes-Silva et al, 2020).

After observing the lack of CENP-W, we aimed to address how its loss has been compensated. By analyzing the structure of bmCENP-T, we report unprecedented changes in the HFD that alleviate the need for stabilization by an interacting partner. We detect a reorientation of α3 in the HFD that brings the helices of the HFE closer to the HFD, so that they occupy the position of CENP-W. In this arrangement, the extension might stabilize the hydrophobic residues of the HFD, thereby acquiring the function of CENP-W, while still retaining its conserved role in interacting with CENP-HIKM. Interestingly, this structural rearrangement of the HFD was present in CENP-T from all insect orders that lost CENP-W. In line with these observations, we find lepidopteran CENP-T to be a stable monomer in solution that retained its ability to bind DNA without the need of an interacting partner.

## Results

### Lepidopteran CENP-T is a soluble monomeric protein independent of CENP-W

To discover potential HFD partners of CENP-T that we might have missed in our previous analyses, we ectopically expressed the C'-terminal fragment of *Spodoptera frugiperda* CENP-T (sfCENP-T) that includes the HFD and the two-helical histone fold extension (HFE) in SF9 cells, a lepidopteran cell line derived from *S. frugiperda*. We detected the sfCENP-T fragment mainly in the supernatant (soluble) fraction of the cell lysate after centrifugation by western blot analysis (Fig. 1B). The insoluble pellet fraction had

relatively lower levels of the protein. For HFD proteins to remain soluble, they must form dimers—either by homo-dimerizing with themselves, as seen in archaeal histones (Stevens et al, 2020; Hocher and Warnecke, 2024; Mattiroli et al, 2017) or by heterodimerizing with another HFD protein, as it is the case for the vertebrate CENP-T and CENP-W. To distinguish between these two possibilities, we purified the sfCENP-T fragment over size-exclusion chromatography (SEC) and performed Multi Angle Light Scattering (MALS) analyses to determine the molecular weight of the potential complex that it might form. Analysis of the fractions enriched for the purified protein by SDS-PAGE did not reveal any coeluting proteins (Figs. 1C and EV1A), supporting the absence of a previously undetectable ortholog of CENP-W or the formation of a complex with another histone-fold protein. SEC-MALS analyses further revealed that the sfCENP-T fragment was a monomer in solution, hitherto unknown for any HFD-containing protein (Fig. 1D). The CENP-T-HFD from the second lepidopteran species *B. mori* was also found to be a soluble monomeric protein, further strengthening our observations (Fig. EV1B,C), These findings corroborate that the HFD in lepidopteran CENP-T folds independently of an interacting partner and is a monomer in solution.

### An acquired role of the histone fold extension in stabilizing CENP-T

To understand the molecular basis of this stability, we used AlphaFold to model the structure of full-length CENP-T from *B. mori*, the most established lepidopteran model organism in which CENP-T was functionally characterized (Fig. 2A). While the N-terminus was largely disordered, the C-terminal HFD and HFE were highly similar across different AlphaFold models and were generated with high confidence (pLDDT>90) (Fig. EV2A–C). We also modeled the sfCENP-T, which showed a high structural similarity to the predicted bmCENP-T structure (Fig. EV2D). A comparison of the best-ranked model of bmCENP-T with the structure of the canonical CENP-TW heterodimer (as reported for chicken) revealed two striking rearrangements. First, the third helix (α3) of the HFD is positioned parallel to the second helix (α2) in bmCENP-T (that we refer to as monomeric HFD hereafter) as opposed to its position perpendicular to the α2 in the conventional HFD. Second, this altered position of α3 enables the HFE to be closer to the HFD compared to the canonical CENP-T (Fig. 2A,B).

The position of α3 in the canonical fold allows access to CENP-W such that it is trapped between the helices of the HFD and the two-helical extension. This configuration creates two sets of hydrophobic surfaces—the central helix of CENP-W and the buried parts of the helices of HFD and the two-helical extension. This arrangement facilitates the formation of a hydrophobic core that stabilizes the heterodimer (Fig. 2C). In the case of lepidopteran CENP-T, the rearrangements in the monomeric HFD bring the two helices of the HFE in the position otherwise occupied by CENP-W. The proximity between the hydrophobic residues of the HFE and those of α1–2 of the HFD enables the formation of a hydrophobic core, making the protein fragment stable and soluble independently of CENP-W (Figs. 2D and EV2E). This is further supported by the conservation of hydrophobic residues across the HFD and the HFE across insect CENP-T sequences (Fig. EV2F). To rule out oligomerization in the presence of the N-terminus of CENP-T, we predicted a homodimeric structure of full-length bmCENP-T (Fig. EV2G). The predicted structures were of poor confidence scores, and the dimerization interface in this structure would occlude the interaction between CENP-T and CENP-HIKM, making it physiologically improbable.

To validate the structure of the altered HFD, we initially attempted to solve the structure of the BmCENP-T HFD by crystallography, which was unsuccessful (see Supplemental Information). We therefore used Crosslinking Mass Spectrometry (CLMS) to identify intra-protein contacts and assessed the distance between these pairs in the AlphaFold model. We used the zero-length crosslinker EDC to identify crosslinks between the HFD and HFE that are in proximity in the predicted structure of the altered HFD (10–13 Å, within the EDC crosslinking range). If the actual position of HFE is like that of a canonical fold, it would be beyond the crosslinking range of EDC, making it an ideal crosslinker to distinguish between these two conformations (Fig. 2E,F; Dataset EV1). MS analyses of the crosslinked protein revealed that 6 of the 10 crosslinks involving a K/D/E residue in HFE were with the residues in α2-3 HFD. In the AlphaFold model, these residue pairs were found to be 9.2–15.7 Å apart, indicating that the protein adopts a further compacted structure in solution. This validates the proximity of the HFE and its role in stabilizing the HFD in the predicted structure. Overall, these results provide a molecular basis for the monomeric nature of bmCENP-T HFD fragment

## The repositioned histone fold extension occludes the interaction with CENP-W

From the structural predictions of bmCENP-T, we inferred that the HFE occupies the channel conventionally occupied by CENP-W. To further explore the impact of the altered organization of the bmCENP-T C-terminus on its interaction with CENP-W, we used a yeast two-hybrid assay (Fig. 2G). We used the interaction between the human CENP-T C-terminus (HFD and the two-helical extension) and human CENP-W as a positive control, because both proteins have a canonical HFD and are known interacting partners (Fig. 2G). We were also able to detect the interaction between the canonical HFD pairs human CENP-T and *Saccharomyces cerevisiae* CENP-W (scCENP-W), suggesting that the assay was sensitive enough to detect interaction between proteins with compatible folds but from phylogenetically distant hosts (Fig. 2G).

However, we could not detect an interaction between the bmCENP-T C-terminus (monomeric HFD and two-helical extension) with scCENP-W (Figs. 2G and EV2H). AlphaFold Multimer predictions also suggested an incompatibility between bmCENP-T and scCENP-W (Fig. EV2I). Collectively, these observations concur with our model that the structural rearrangements in bmCENP-T prevent its interaction with CENP-W.

## The DNA-binding ability of CENP-T is conserved in *B. mori*

Given the divergence of *B. mori* CENP-T from the canonical CENP-T structure, we next tested if the DNA-binding ability of *B. mori* CENP-T was retained using electrophoretic mobility shift assays. Like the canonical CENP-TW complex (Nishino et al, 2012), the bmCENP-T C-terminal fragment [bmCENP-T (wt)] was able to shift DNA, suggesting that the monomeric HFD retained the DNA binding ability. We observe maximum DNA-protein complex formation by incubating a 12 bp template DNA with 5x-molar excess of protein (Figs. 3A and EV3A).

To further dissect the regions of bmCENP-T that bind to DNA, we identified positively charged regions on the protein surface (Fig. 3B). By comparing these with the residues known to contact DNA in human CENP-T (Nishino et al, 2012), we assigned three patches that might mediate DNA interactions along the bmCENP-T fragment, namely patch 1 (located at α1 and sequences upstream of it, with residues K895, R896, K907, R908, K911), patch 2 (with residues K928, R963 located across α2-3), and patch 3 (with residues R983, R986 located at the C' extension) (Fig. 3B) These residues show a high degree of conservation across lepidopteran CENP-T sequences (Fig. EV3B). To assess the relative contribution of each of these patches in DNA binding, we mutated all Arg and Lys residues in each of these patches to Ala and Ser, respectively. The SEC elution profile of bmCENP-T[patch1] and bmCENP-T[patch2] mutant proteins was similar to the bmCENP-T, indicating that these mutations do not affect the overall folding of the protein (Fig. EV3C,D). Early elution of CENP-T[patch3] mutant in the SEC indicated deviation from the WT structure and hence this protein was not considered for subsequent analysis (Fig. EV3E).

Among the mutants tested, we observed a loss in DNA binding only for the bmCENP-T[patch1] mutant, suggesting a critical role for patch 1 residues in contacting DNA (Fig. 3C). The bmCENP-T[patch2] mutations had no effect on DNA binding (Fig. 3D). To dissect the DNA-patch 1 interactions further, we sub-categorized these residues into patch 1.1, which includes residues upstream of α1 (K895, R896) and patch 1.2, which includes residues within α1 (K907, R908, K911). Both mutants failed to bind DNA showing that the entire patch 1 plays an essential role in binding DNA (Fig. EV3F,G). These studies demonstrate that the DNA-binding ability of CENP-T is conserved despite the changes in the histone fold.

To understand the functional relevance of patch1 residues for mitotic progression in cells, we tested whether ectopically expressed bmCENP-T[WT] or bmCENP-T[patch1] bearing the same substitutions tested in our electrophoretic mobility shift assays can rescue mitotic defects observed upon endogenous CENP-T depletion in a *B. mori*-derived cell line BmN4-Sid1 (Kobayashi et al, 2012). As previously described (Cortes-Silva et al, 2020), RNAi-mediated depletion of endogenous bmCENP-T resulted in mitotic arrest, seen in the form

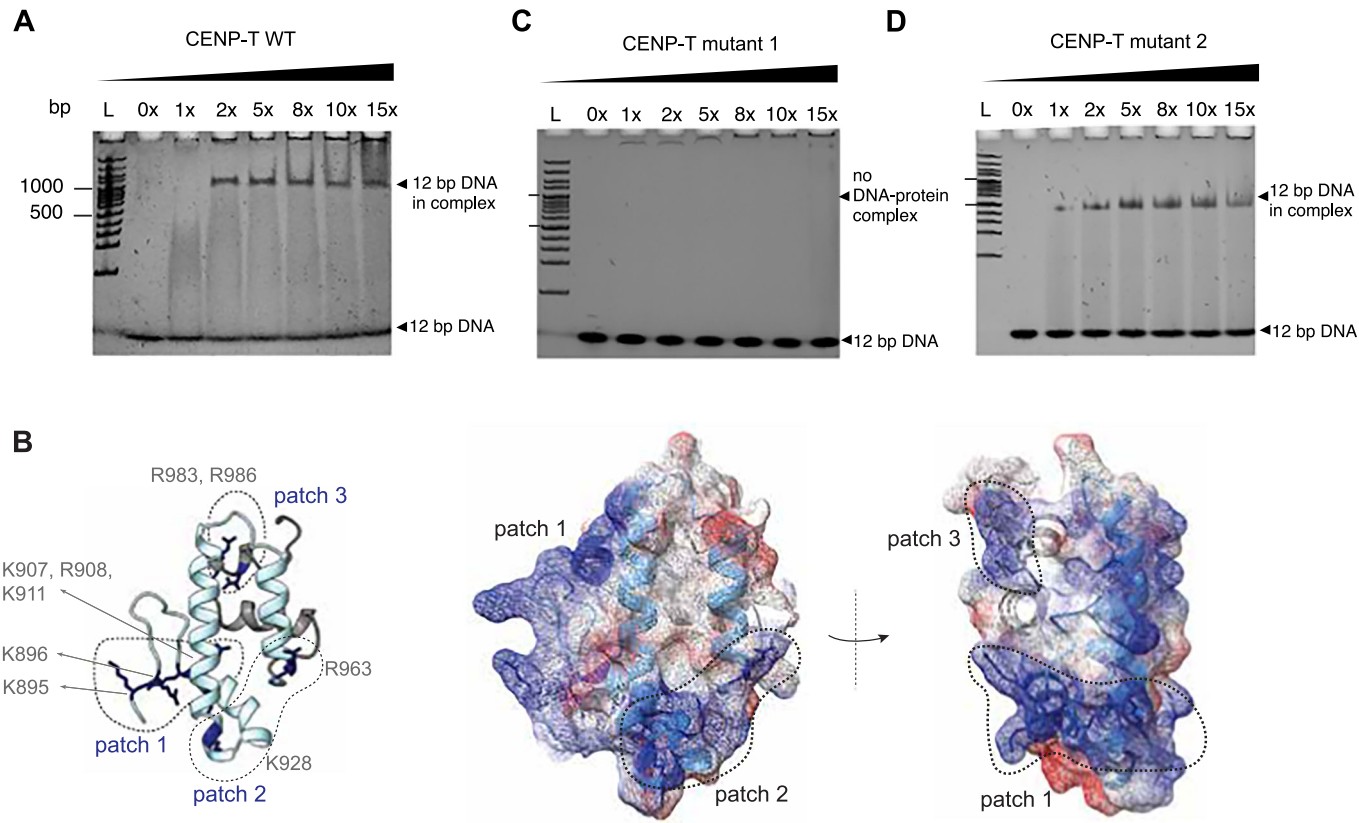

**Figure 3. The DNA-binding ability of CENP-T is conserved in *B. mori*.**

(A) 5% Native PAGE gels of DNA titrations using different molar ratios of bmCENP-T[894-1016]-WT with a 12 bp DNA in 100 mM NaCl. Lane L: 100 bp DNA marker. (B) The positively charged surfaces formed by the conserved lysines and arginine residues in patches 1–3 are highlighted. The position of these residues and their side chains (dark blue) is shown in the ribbon model on the left. The surface charge distribution of the same model is depicted on the right. The front and side views are shown for this model with the positively and negatively charged surfaces colored blue and red, respectively. (C, D) Native PAGE gels of DNA titration using different molar ratios of bmCENP-T[894-1016] patch 1 mutant with the substitutions K895A, R896A, K907A, R908S, and K911A (C) and bmCENP-T[894-1016] patch 2 mutant with the substitutions R928S and R963S (D). The assay was performed with a 12 bp DNA fragment and indicated amounts of protein at 100 mM NaCl. Lane L: 100 bp DNA marker. Source data are available online for this figure.

---

of congression and metaphase alignment defects (Fig. EV4A–C). As expected, these defects were rescued by the expression of an RNAi-resistant version of bmCENP-T[WT]. Notably, we also find the RNAi-resistant version of bmCENP-T[patch1] to have a similar effect, suggesting that the DNA binding ability through these residues in bmCENP-T is not essential for accurate mitotic progression, at least in a *B. mori* cell line.

## The repositioned histone fold extension still mediates CENP-T: CENP-HIKM interaction

Besides DNA binding, the HFE of CENP-T is also essential for its kinetochore localization through its interaction with CENP-K of the CENP–HIKM complex (McKinley et al, 2015; Pekgöz Altunkaya et al, 2016; Nishino et al, 2012; Basilico et al, 2014). With the observed rearrangements in lepidopteran CENP-T, we predicted the interface of this interaction using AlphaFold multimer (Fig. EV4D) and tested for its functional conservation as follows. In a stable Sf9 cell line that contains a genomically integrated *LacO* array, we ectopically expressed LacI-GFP-sfCENP-

I in tandem with either 3xFLAG-tagged sfCENP-T[WT] or sfCENP-T[Δtail], wherein the potential CENP-K-interacting part of the protein was deleted. The ability of sfCENP-T[WT] to interact with CENP-HIKM was supported by its localization to the *LacO* foci bound by LacI-GFP-sfCENP-I (Fig. EV4D,E). A significantly diminished localization of sfCENP-T[Δtail] to the *LacO* foci suggested that the HFE of CENP-T still retained its role in mediating CENP-T and CENP–HIKM interaction (Fig. EV4E). We next studied the functional relevance of this association by testing the kinetochore localization of CENP-T[Δtail] and its role in mitotic progression (Fig. EV4A–C). A RNAi-resistant version of 3xFLAG-tagged bmCENP-T[Δtail] was expressed in BmN4-Sid1 cells, wherein the endogenous CENP-T was depleted. Unlike bmCENP-T[WT], cells transfected with bmCENP-T[Δtail] accumulated in mitosis and showed defective chromosome segregation comparable to the levels observed upon endogenous CENP-T depletion (Fig. EV4A). The lack of a typical kinetochore localization of bmCENP-T[Δtail] is in line with this observation (Fig. EV4C). Together, our results suggest an essential role for the HFE in the localization and function of CENP-T in Lepidoptera.

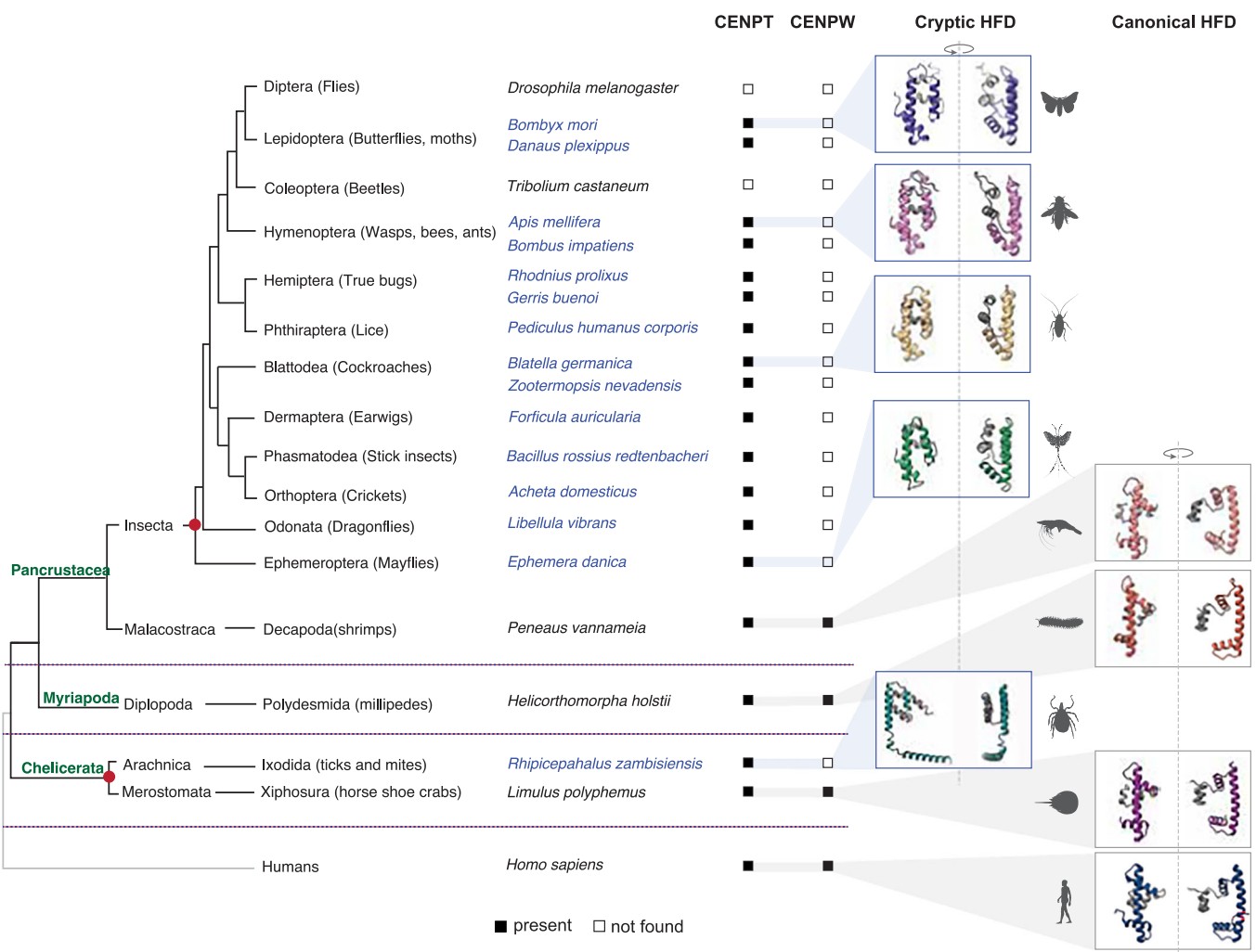

**Figure 4. Independent evolution of the monomeric HFD in Arthropods.**

A cladogram depicting the three major arthropod classes Chelicerata, Myriapoda, and Pancrustacea, with humans as an outgroup. Species representative of each of these classes are mentioned along with their respective subclass and order. The presence/absence of indicated proteins in each species is marked by filled and empty boxes, respectively. For protein IDs, see Dataset EV2. The configuration of the CENP-T HFD was assessed using AlphaFold for one species in each taxonomic order. Species with a monomeric CENP-T HFD are colored in blue, and the ones with a canonical CENP-T HFD are colored black. The cartoons on the right are two different views of the predicted model to highlight the proximity of the C' extension. The red circles mark the two independent origins of the monomeric HFD identified in this study.

## The origin of the monomeric HFD can be traced to the last common ancestor of insects

Having demonstrated that lepidopteran CENP-T can fold stably and associate with DNA in the absence of CENP-W, we expanded the analysis of CENP-T structures to other insect orders. Except for the order Coleoptera (beetles), CENP-T homologs are present in all other insect orders. However, no insect CENP-W homolog has been identified to date. Based on the AlphaFold predictions of CENP-T structures of each insect order, the monomeric HFD is present in all insect orders, including Ephemeroptera, an early branching insect order (Fig. 4). This conservation places the evolution of the monomeric HFD to the last common ancestor of insects, which might have co-occurred with the loss of CENP-W.

To understand the sequence variations associated with the evolution of this monomeric fold, we measured the amino acid substitutions across the HFD using Shannon's diversity index. Due to the evolutionary conservation of HFDs, we expect a minimization of Shannon's diversity score (entropy) for the residues in this domain, which we observed when conventional CENP-T HFD sequences from vertebrates were analyzed (Fig. EV5A). In the case of insects, we see a reduction in sequence diversity only for the residues of α1 and the N' half of α2. For residues corresponding to regions folded differently in the monomeric fold, we observe a marked increase in sequence diversity, hinting at an ongoing evolutionary process or lower constraints on the amino acid identity in this region (Fig. EV5B). While we did not detect any sequence signature specific to the altered fold, we find the hydrophobicity of several amino acids in the HFE and HFD conserved (Fig. EV2F).

We extended our search beyond insects to include other arthropod clades and were able to identify additional CENP-T homologs in species belonging to Xiphosura (ex: horseshoe crabs), Ixodida (ex: ticks), Polydesmida (ex: millipedes), and Decapoda (ex: shrimps). We generated AlphaFold models for each of these CENP-T homologs. In *Limulus polyphemus* (horseshoe crab), *Helicorthomorpha holstii* (millipede), and *Paneaus vannameia* (shrimp), CENP-T adopts a canonical HFD structure. Consistent with this, we also identified homologs of CENP-W in each of these organisms. In contrast, in the tick *Rhiphicephalus zambisiensis*, CENP-T was predicted to adopt the altered HFD structure with proximal HFE as described in this study (Fig. 4). We failed to detect a CENP-W homolog in this species. Intriguingly, we could also detect other configurations of the HFD in related tick species where CENP-W homologs remain undetected. Notable example includes a configuration wherein the three helices of the HFD adopt a canonical structure, but the space occupied by CENP-W-α2 now hosts an alpha helix from the N' of CENP-T itself (Fig. EV5C–E). Like the monomeric HFD, this configuration could also stabilize CENP-T to make it an independently folding protein. This might mark a second independent origin of a monomeric HFD within the arthropods, highlighting the adaptability of CENP-T proteins to bottlenecks, such as the loss of an interacting partner.

# Discussion

In this study, we report an unprecedented rearrangement in the structure of the HFD in insect CENP-T orthologs. This structural change alleviates the need for an HFD partner and renders the protein soluble as a monomer. This is highly unusual for the family of HFD proteins known to be obligate dimers across the tree of life. While the HFD shows limited sequence conservation, both the tertiary structure and its functional organization as a dimer stabilized by a handshake interaction with an HFD-partner are conserved across the diverse range of proteins with this domain. The identification of a variant in such a highly conserved and optimized protein module offers a unique window to understand the evolution of protein structure.

We attribute the stability of monomeric CENP-T to the two-helical extension at the C-terminus, which is in proximity to the histone fold due to the repositioning of the α3 in the monomeric HFD as compared to a canonical CENP-T (Nishino et al, 2012). In canonical CENP-T, the HFD is critical for the interaction with CENP-W and DNA, while the two-helical extension is critical for the interaction with CENP-K of the CENP-HIKM complex (Hori et al, 2008; Nishino et al, 2012; Pesenti et al, 2022; Yatskevich et al, 2022; Yan et al, 2019; McKinley et al, 2015; Hinshaw and Harrison, 2020; Zhang et al, 2020). Despite the observed rearrangements in the HFD, bmCENP-T still binds DNA in vitro. This interaction is dependent on residues in the α1 of the HFD and the region upstream of it, both of which are also the regions of CENP-T that contact DNA in human and yeast CCAN/inner kinetochore complexes (Fig. EV5F) (Yan et al, 2019; Yatskevich et al, 2022; Pesenti et al, 2022). In chicken and human cells, localization of CENP-T and the formation of functional kinetochores were dependent on the ability of CENP-T to bind DNA (McKinley

et al, 2015; Nishino et al, 2012). In contrast, we could detect the localization of CENP-T^patch1 to mitotic chromosomes and its ability to rescue defects caused by the depletion of endogenous CENP-T. It is plausible that CENP-T^patch1 retains function in ambient growth conditions used in our assay, and additional stresses/conditions need to be assayed to detect sensitivities of this mutant form. Deviation from a conserved dependency on DNA binding could also be indicative of functional divergence of CENP-T in Lepidoptera. Whether this can be attributed to the presence of multiple kinetochores along the chromosome (holocentric architecture) remains to be investigated.

The sequence of the two-helical extension that mediates CENP-T recruitment to the kinetochores by interacting with the CENP-HIKM complex in vertebrates is also conserved in Lepidoptera. The loss of CENP-T localization upon depletion of CENP-HIKM subunits in *B. mori* cells, or upon deletion of the C-terminal tail in the HFE in both *B. mori* and *S. frugiperda* validates that the function of the two-helical extension remains conserved (Cortes-Silva et al, 2020). We propose that the stabilization of the monomeric HFD is an acquired, second function of the two-helical extension in Lepidoptera. Taken together, our study highlights a structural innovation in CENP-T that supports its stability even in the absence of CENP-W, while leaving the other functional associations of CENP-T unperturbed.

The presence of a C-terminal two-helical extension that is unique to CENP-T orthologs might make CENP-T uniquely poised among HFD proteins to evolve the ability to become soluble as a monomer. That said, the HFE is highly conserved in CENP-T from yeast to humans irrespective of the structure of the HFD or the availability of CENP-W (Hinshaw and Harrison, 2020; Zhang et al, 2020; Cortes-Silva et al, 2020; Pekgöz Altunkaya et al, 2016). This leads us to hypothesize about the evolutionary drivers and constraints that may have triggered the origin of a monomeric fold in the last common ancestor of insects, and not in other lineages. The conserved HFD enables binding to CENP-W, which is necessary for the recruitment of CENP-SX as part of a tetrameric CENP-TWSX complex (Takeuchi et al, 2014; Nishino et al, 2012). While CENP-SX is not essential for cell viability, depletion of CENP-SX subunits in vertebrates leads to defective outer kinetochore formation and defects in mitotic progression (Amano et al, 2009; Nishino et al, 2012). The contributions of CENP-SX to outer kinetochore assembly might act as a functional constraint to retain these subunits at the kinetochore and in turn preserve a canonical CENP-T HFD in vertebrates and fungi. In insects, CENP-SX subunits are either lost or not part of the kinetochore (Drinnenberg et al, 2014; Cortes-Silva et al, 2020). Their absence at the insect kinetochore renders the formation of a heterodimer between CENP-T and CENP-W redundant. Under these circumstances, the evolution of an altered histone fold might be advantageous because it removes the dependency on CENP-W. This optimization gains significance in Lepidopteran systems that lost the classical CENP-C-based link between the inner and outer kinetochore, and CENP-T remains to be the only direct linker (Cortes-Silva et al, 2020). Finally, the potential recurrence of the monomeric HFD of CENP-T in other arthropods raises the intriguing question whether similar changes might have occurred in additional eukaryotic lineages.

# Methods

**Reagents and tools table**

| Reagent/resource | Reference or source | Identifier or catalog number |
|---|---|---|
| **Experimental models** | | |
| BmN4 | ATCC | Cat#CRL-8910; RRID: CVCL_Z633 |
| BmN4-SID1 | Kobayashi et al (2012) | RRID:CVCL_Z091 |
| Sf9 | GIBCO | Cat#12659017 |
| Sf9-LacO | Cortes-Silva et al (2020) | N/A |
| _S. cerevisiae_ strain AH109 _MATa, trp1-901, leu2-3, 112, ura3-52, his3-200, gal4Δ, gal80Δ, LYS2 : : GAL1$_{UAS}$-GAL1$_{TATA}$-HIS3, GAL2$_{UAS}$-GAL2$_{TATA}$-ADE2, URA3 : : MEL1$_{UAS}$-MEL1$_{TATA}$-lacZ_ | James et al (1996) | |
| **Bacterial and viral strains** | | |
| _E. coli_ Bl21DE30 plys pRare | Sigma | 71400 |
| _SfCENPT_ $^{HFD}$ baculovirus | This study | N/A |
| **Antibodies** | | |
| Mouse monoclonal Anti-FLAG M2 antibody | Sigma | Cat#F1804; RRID: AB_262044 |
| Rabbit polyclonal anti-CENP-T (rabbit 045) | Cortes-Silva et al (2020) | N/A |
| Rabbit polyclonal anti-Dsn1 (rabbit 1615031) | Cortes-Silva et al (2020) | N/A |
| Mouse monoclonal anti-6xHis | Sigma | Cat#ab18184; RRID:AB_444306 |
| Goat monoclonal IRDye 680RD anti-Rabbit IgG LI-COR | | Cat#926-68071; RRID:AB_10956166 |
| Donkey monoclonal IRDye 800CW anti-mouse IgG LI-COR | | Cat#926-32212; RRID:AB_621847 |
| Mouse monoclonal anti-FLAG M2 beads | Sigma | Cat#M8823; RRID:AB_2637089 |
| Mouse monoclonal anti-α-tubulin Alexa Fluor 488 | Thermo Fisher Scientific | Cat#53-4502-80; RRID:AB_1210526 |
| Mouse monoclonal anti-FLAG M2 | Sigma | Cat#F1804; RRID:AB_262044 |
| Rat monoclonal anti-phospho Histone H3-Ser10 | Sigma | Cat#MABE939 |
| Goat polyclonal anti-rabbit IgG Alexa Fluor 568 | Thermo Fisher Scientific | Cat#A-11011; RRID:AB_143157 |
| Goat polyclonal anti-rat IgG Alexa Fluor 568 | Thermo Fisher Scientific | Cat#A-11077; RRID:AB_2534121 |
| Goat polyclonal anti-rat IgG Alexa Fluor 488 | Thermo Fisher Scientific | Cat#A-11006; RRID:AB_2534074 |
| Goat polyclonal anti-mouse IgG Alexa Fluor 488 | Thermo Fisher Scientific | Cat#A-11029; RRID:AB_2534088 |
| Goat polyclonal anti-mouse IgG Alexa Fluor 568 | Thermo Fisher Scientific | Cat#A-11004; RRID:AB_2534072 |
| Goat polyclonal anti-rat IgG Alexa Fluor 633 | Thermo Fisher Scientific | Cat#A-21094; RRID:AB_2535749 |
| **Chemicals, enzymes, and other reagents** | | |
| X-tremeGENE™ HP DNA Transfection Reagent | Merck | Cat# XTGHP-RO |
| Complete Protease Inhibitor Cocktail | Roche | Cat#11697498001 |
| Bolt 4-12% Bis-Tris Plus denaturing gels | Invitrogen | Cat#NW04120BOX |
| Novex 16% Tris Glycine Precast Gels | Invitrogen | Cat#XP00162BOX |
| Magnetic Dynabeads Protein A | Invitrogen | Cat#10002D |
| Benzonase | In-house | |
| 4-20% Tris glycine gels | Invitrogen | Cat#XP04200BOX |
| Instant_Blue_ | Sigma | Cat#ISB1L |
| PVDF membrane | Bio-Rad | Cat#170-4272 |
| Odyssey Blocking buffer | LI-COR | Cat#927-50000 |
| DTT, dithiothreitol | Euromedex | Cat#EU0006-B |

| Reagent/resource | Reference or source | Identifier or catalog number |
|---|---|---|
| DAPI | Sigma | Cat#D9542 |
| Vectashield Antifade Mounting Medium | Vector Laboratories | Cat# H-1000; RRID:AB_2336789 |
| EDC | Thermo Scientific | A35391 |
| Sulfo-NHS | Thermo Scientific | A39269 |
| Isopropyl-b-D-1-thiogalactopyranoside | Sigma | I5502 |
| Protino Ni-NTA agarose beads | MN | REF 745400.100 |
| Fugene HD transfection reagent | Promega | E2311 |
| HiLoad 16/600 Superdex 75pg | Cytiva/GE healthcare | 28989333 |
| Gibson Assembly mastermix | New England Biolabs | E2611L |
| Restriction enzymes | New England Biolabs | |
| **Software and algorithms** | | |
| MAFFT | Katoh and Standley, 2013 | https://mafft.cbrc.jp/alignment/software/ |
| Jalview | Waterhouse et al 2009 | https://www.jalview.org/ |
| HMMER webserver | Potter et al, 2018 | http://www.ebi.ac.uk/Tools/hmmer |
| HHpred version 3.2.0 | Zimmermann et al, 2018 | https://toolkit.tuebingen.mpg.de/tools/hhpred |
| Fiji | Schindelin et al, 2012 | http://fiji.sc/ |
| Prism version 8.12 for Mac | GraphPad Software | https://www.graphpad.com/scientific-software/prism/ |
| NCBI Blast suite | | https://blast.ncbi.nlm.nih.gov/Blast.cgi?PROGRAM=blastp&PAGE_TYPE=BlastSearch&LINK_LOC=blasthome |
| **Other** | | |
| *S. frugiperda* "corn strain" annotation | Gouin et al, 2017 | https://bipaa.genouest.org/sp/spodoptera_frugiperda_pub/ |
| **Oligonucleotides and other sequence-based reagents** | | |
| **Identifier** | **Sequence (5'-3')** | |
| SpoCENPT-HFD-His-BamHI-F | GCGCAGGATCCATGGGCAGCA GCCATCACCATCAT CACCACAGCCAGCCGATG TTCAAAGTACCAAACAAACCA | |
| SpoCENPT-HFD-PstI-R | GCGCA CTG CAG TCA TCCTTGTA CATTGTTCCCTCTTAG | |
| CENP-T $_{bm}^{894-1016}$ FWD | CAGGGGCCCCTGGGATCC | |
| CENP-T $_{bm}^{894-1016}$ REV | TTAATTAACTCGAGCGGC | |
| SR107-2 bmT-F | GACAGGATCCGAAAAGA TATCAACCAAAGAATGC | |
| SR108 bmT-R | GACACTGCAGCTACGCA TGTACCGCATG | |
| attb1-hHFD+ext-F | GGG GAC AAG TTT GTA CAA AAA AGC AGG CTC CGG ACT GAG CCA CTA TGT GAA AC | |
| attb2-hHFD+ext-R | GGG GAC CAC TTT GTA CAA GAA AGC TGG GTC CTA CTG GGC AGG GAA GAC AG | |
| attb1-BmHFD+ext-F | GGG GAC AAG TTT GTA CAA AAA AGC AGG CTC CAA GAG ACT GTA CAA ATA TTT G | |
| attb2-BmHFD+ext-R | GGG GAC CAC TTT GTA CAA GAA AGC TGG GTC CTA CGC ATG TAC CGC ATG CCC AC | |
| attb1-ScCenpW-F | GGG GAC AAG TTT GTA CAA AAA AGC AGG CTC CAT GGA TAC GGA AGC ATT GGC | |

| Reagent/resource | Reference or source | Identifier or catalog number |
|---|---|---|
| attb2-ScCenpW-R | GGG GAC CAC TTT GTA CAA GAA AGC<br>TGG GTC CTA TTG ACT ATC AGA AAG AGC CTG | |
| attB_hCenpW-F | GGGGACAAGTTTGTACAAA<br>AAAGCAGGCTCCATGGCGCTGTCGACCATAGTC | |
| attB_hCenpW-R | GGGGACCACTTTGTACAAGAAAGC<br>TGGGTCCTAACCTCTGCTCTTCTTTAGAATTAC | |
| GC129-F | GAGGGCCACCATGGATAGATCCGGAAAGC | |
| GC130-R | TCGGCGTCGGCTATTCCTTTGCCCTCGG | |
| Tubulin_prom_Fcloning | TGTATCTTATCATGTCTGGATCTTTCAGTCGTGTAGTTG | |
| Tubulin_prom_Rcloning | CTCGGTACCAAGCTTTAAATTTTGATTTGAGTTTTTTTCTATGC | |
| GC127-F | AAAGGAATAGCCGACGCCGACCAACACC | |
| GC128-R | ATCTATCCATGGTGGCCCTCCTATAGTGAGTC | |
| pIVZ5_Hyg_GFP_F | ATTTAAAGCTTGGTACCG | |
| pIVZ5_Hyg_GFP_R | TCCAGACATGATAAGATACATTG | |
| GC161-F | catgtctggaCCACCTACTTTGAGATATG | |
| GC162-R | aagtaggtggTCCAGACATGATAAGATACATTG | |
| PL075-F | TTGGTACCGAGCTCGGATCCATGGACTATAAAGACCATGACG | |
| PL078-R | gcgggccctctagactcgagctacgcatgtaccgcatg | |
| GC104-F | CTCGAGTCTAGAGGGCCC | |
| PL008-R | GGATCCGAGCTCGGTACC | |
| VA129-F | ATGATGATGATAAGGCCGCAATGCCAAGTTCAAAGATACC | |
| GC244-R | gcgggccctctagactcgagTTACGCATGTACCGCATG | |
| VA136-F | CTCGAGTCTAGAGGGCCC | |
| PL022-R | TGCGGCCTTATCATCATCATC | |
| PL098-F | TTGGTACCGAGCTCGGATCCATGCCAAGTTCAAAGATAC | |
| SR142 | ctgtcattgttgtaaggcg | |
| SR140 | ttacaacaatgacagaggaaataatactagtaacaaggatc | |
| SR144 | gggccctctagactcgagttacgcatgtaccgcatgc | |
| PL111-F | ctcgagtctagagggccc | |
| GC238-F | gccgcagactataaagaccatgacg | |
| GC240-R | caccttgacgcgaacttctctc | |
| SR155.3 | GACTCTCGAGTTATTTAACCCTTATTTCTCGGGGC | |

## Generation of vectors for the purification of CENP-T-HFD fragment from S. frugiperda and B. mori

The sequences of the primers mentioned below can be found in the Reagents and Tools table.

The 6xHis-tagged C-terminal fragment (1247–1314 aa) of the S. frugiperda CENP-T (sfCENP-T$^{1247-1314}$) was amplified from cDNA using the primer pairs SpoCENP-T-HFD-His-BamHI-F/ Spo-CENP-T-HFD-PstI-R and cloned into BamHI and PstI sites of the pFastBac-Dual vector by restriction digestion and ligation. The resulting plasmid was named pFastBacDual-SpodoCENP-T-HFD

The bmCENP-T-HFD fragment (894–1016 aa) was amplified from bm genomic cDNA using the primer pairs CENP-T $_{bm}$$^{894-1016}$ FWD and CENP-T $_{bm}$$^{894-1016}$ REV and cloned in frame with the GST

tag at the BamHI–HindIII sites of pGEX-6p vector. The resulting plasmid was named pGEX-CTlong.

To assess the solubility of sfCENP-T-HFD fragment in E. coli, the sfCENP-T-HFD fragment along with the N-terminal 6xHIS tag from the pFastBacDual-SpodoCENP-T-HFD was released by NcoI-PstI digestion and ligated to the same sites of pRSF-Duet vector. The resulting plasmid was named pRSF-sfT.

To assess the solubility of bmCENP-T-HFD in E. coli, the fragment was amplified from pGEX-CTlong using the primer pairs SR107-2 bmT-F/SR108 bmT-R and cloned into the BamHI and PstI sites of pRSFDuet-1 plasmid. The resulting plasmid was named pRSF-bmT.

In each case, the ligation product was used to transform competent DH5α cells. The transformants with pFastBacDual and

pGEX-6p vectors were selected in LB plates supplemented with 100 µg/mL ampicillin, and the transformants with pRSF-Duet1 vector were selected in LB plates supplemented with 30 µg/mL kanamycin. Clones were verified for proper integration by PCR and Sanger sequencing.

### *Spodoptera frugiperda* CENP-T C-terminus expression in SF9 cells, purification, and SEC-MALS analysis

To purify the 6xHis-tagged sfCENP-T[1247–1314] fragment a recombinant baculovirus was generated to infect SF9 cells from 400 ml at a density of $\sim 1 \times 10^6$ cells/ml. After 62 h of infection, cells were harvested (density $\sim 1 \times 10^6$ cells/ml) and the pellet was washed using wash buffer (20 mM Tris pH 8, 200 mM NaCl, 25 mM imidazole). Cells in 15 ml of wash buffer were lysed after the addition of 1 Pierce™ Protease Inhibitor Tablet EDTA-free (Antiprotease 1 tablet per 200 mL, Thermo Scientific)and 200 µL Triton-X-100 to rotate for 30 min at 4 °C. Cells were then lysed by sonication (15 × 20 s at 25% amplitude with 30 s rest between) and spun down at 20,000×g for 2 h at 4 °C to separate the soluble and insoluble fractions. Proteins from the soluble fraction were purified over 3 ml Protino Ni-NTA Agarose columns (Sigma-Aldrich) and eluted using 4 ml Elution buffer (20 mM Tris pH 8, 200 mM NaCl, 300 mM imidazole). Eluates were centrifuged at 16,000×g for 10 min at 4 °C to remove aggregates. Gel filtration chromatography in gel fitration buffer (20 mM Tris pH 8.0, 300 mM NaCl, 5% glycerol, 1 mM DTT) was performed using HiLoad™ 16/600 Superdex® 75 pg 120 mL analytical column (GE Healthcare) injected with 3.5 ml of the eluate. For Fig. 1C and EV1A SEC fractions were visualized on SDS-PAGE stained with Coomassie Blue.

For the SEC-MALS analyses, CENP-T fragments were purified in the same way but from 800 ml cultures. After Ni-NTA pulldown, eluates were purified over Heparin Columns in 200 mL of 20 mM Tris pH 8.0 binding buffer, eluted using salt gradients up to 1.5 M NaCl to elute the protein. Fractions corresponding to the recombinant protein were pooled and concentrated using Amicon™ 3000 MWCO Ultra-15 Centrifugal Filter Units (Sigma-Aldrich Merck). Gel filtration chromatography in gel filtration buffer (20 mM Tris pH 8.0, 300 mM NaCl, 5% glycerol, 1 mM DTT) was performed using HiLoad™ 16/600 Superdex® 75 pg 120 mL analytical column (GE Healthcare) injected with 4.5 ml of the eluate. Fractions corresponding to the recombinant protein were pooled and concentrated using Amicon™ 3000 MWCO Ultra-15 Centrifugal Filter Units (Sigma-Aldrich Merck). For SEC-MALS analysis, the sample (~241.1 µg/mL in ~600 µL) were injected in a Superdex 200 10/300 Increase (Cytiva) previously equilibrated in the corresponding buffer, and developed at 0.5 mL/min. Data collection was performed every 0.5 s with a Treos static light scattering detector and a t-Rex refractometer (both from Wyatt Technologies). The concentration and molecular mass of each data point were calculated with the software Astra 6.1.7 (Wyatt Technologies).

### Purification of bmCENP-T-HFD fragment

#### Induction, protein purification, and SEC-MALS
BmCENP-T-HFD was cloned into pGEX-6P vector with a GST tag and PreScission HRV 3 C protease cleavage site on the N-terminal

and expressed using Rosetta 2 BL21 (DE3) Competent Cells® overnight at 18 °C in the presence of 50 µg/mL Amp and 34 µg/mL Cam. The cells were harvested by centrifugation and resuspended in 20 mL of lysis buffer (50 mM Tris pH 7.5, 500 mM NaCl, 5 mM 2-mercaptoethanol, 5% Glycerol, and protease inhibitors (Pierce™ Protease Inhibitor Tablet EDTA free). The cells were lysed by sonication (Branson 550 sonicator) and the lysate was clarified by spinning at 40,000×g for 60 min. BmCENP-T-HFD was first purified using GSTrap™ 4B-5 mL column (GE Healthcare) preequilibrated with GST buffer A (50 mM Tris pH 7.5, 500 mM NaCl, 5 mM BME, 5% Glycerol), and the pure protein was eluted using GST buffer A supplemented with 10 mM glutathione (Sigma-Aldrich). The GST-tag was cleaved using PreScission HRV 3C protease (200 units per 20 mg of protein), and the mixture was dialyzed overnight at 4 °C against dialysis buffer containing 50 mM Tris pH 7.5, a final concentration of 300 mM NaCl, and 5 mM β-ME. The Cleaved BmCENP-T-HFD was then concentrated, clarified, and injected on HiLoad™ 16/600 Superdex® 75 pg 120 mL (GE Healthcare) preequilibrated with 10 mM HEPES pH 7.5 and 500 mM NaCl. The protein quality and purity were assessed using 4–20% Mini-PROTEAN® TGX™ Precast SDS-Page Gels (Biorad), and fractions containing pure protein were pooled and concentrated. SEC-MALS analysis was performed as described for sfCENP-T-HFD fragment using a 50 µL protein sample at concentration 1.95 mg/mL that was injected to a Superdex 75 10/300 column (Cytiva) equilibrated with SEC buffer (10 mM HEPES pH 7.5, 500 mM NaCl).

### Crystallization attempts

Fresh and concentrated BmCENP-T-HFD (9 and 15 mg/ml), with and without preincubation with 12 bp double-stranded DNA, was used to set up different commercial crystallization conditions (Morpheus®, SG1™, and JCSG Plus™) using Oryx 4 (Douglas Instruments) in 96-well SWISSCI plates (sitting drop method). Positively associated hits in the screens were optimized in 24 Well Crystallization Plates (Hampton Research) using the hanging drop method, but single crystals of good quality were never obtained.

### DNA-binding studies

DNA studies were performed by first annealing DNA single-strands (ssDNA) together, followed by electrophoretic mobility shift assay (EMSA) of the protein in complex with DNA.

#### DNA sequences used
10 bp DNA FWD (AA368) TAGACAGCTC
10 bp DNA FWD (AA369) GAGCTGTCTA
12 bp DNA FWD (AA357) TAGACAGCTCTA
12 bp DNA REV (AA358) TAGAGCTGTCTA
14 bp DNA FWD (AA370) TAGACAGCTCTAGC
    14 bp DNA FWD (AA371) GCTAGAGCTGTCTA

#### DNA annealing protocol
The annealing of 8 -, 10 -, 11 -, 12 -, and 14 bp single-stranded DNA (ssDNA) to double-stranded DNA (dsDNA) was done by mixing equal molar ratios of the complementary oligonucleotides and slowly decreasing the temperature from 95 °C to the melting temperature (Tm) of the DNA at a rate of 1 °C/min. The

temperature (Tm) was held for 30 min, followed by lowering the temperature to 22 °C using the same rate. The annealing was tested and checked by loading 300 ng of DNA with 5% sucrose (Sigma) on 3% agarose gel. The annealed DNA was concentrated to 13 mg/ml by ethanol precipitation and stored at −20 °C.

### Electrophoretic mobility shift assay (EMSA)
EMSA was done using 5% native PAGE gels. The binding was tested at different salt concentrations using increased ratios of protein to DNA. The samples were mixed and incubated at 4 °C for 30 min before loading 300 ng of DNA in the presence of 8% glycerol on the gel. The gel was run at 150 V for 25 min using PowerPac™ Universal (Bio-Rad), stained in a 3× aqueous GelRed® (Millipore) solution, and scanned using the ChemiDoc™ MP Imaging system (Bio-Rad).

## CLMS analysis

No-weigh EDC crosslinker and Sulfo-NHS (21585, Thermo Fischer) were dissolved in water to make a stock solution of 50 mM and 115 mM, respectively. A molar ratio of protein: EDC: SulfoNHS of 1:10:25, as suggested by the manufacturer, was used for this assay. The protein was dialyzed against 1× PBS with 300 mM NaCl. The crosslinking reaction was performed by mixing the 55 µg of protein with crosslinkers at the above concentration and incubating the mixture at room temperature for 90 min. The reaction was then quenched by adding Tris pH 8.0 to a final concentration of 100 mM and incubating at 35 °C for 10 min. The crosslink reaction products (5 µg) were visualized by SDS-PAGE. For the identification of the crosslinked residues by mass spectrometry, 50 µg of protein was crosslinked by the same method. After quenching, the samples were snap frozen and shipped to EMBL (Heidelberg) for subsequent MS analysis.

For the digestion, 5 mM TCEP, 20 mM CAA, and 1 µg trypsin were added and incubated at 37 °C overnight. The next day, the reaction was stopped by the addition of 1% TFA. Digested peptides were concentrated and desalted using an OASIS® HLB µElution Plate (Waters) according to manufacturer instructions. Crosslinked peptides were enriched using size-exclusion chromatography (https://doi.org/10.1074/mcp.M111.014126). In brief, desalted peptides were reconstituted with SEC buffer (30% (v/v) ACN in 0.1% (v/v) TFA) and fractionated using a Superdex Peptide PC 3.2/30 column (GE) on a 1200 Infinity HPLC system (Agilent) at a flow rate of 0.05 ml/min. Fractions eluting between 50 and 70 µl were evaporated to dryness and reconstituted in 30 µl 4% (v/v) ACN in 1% (v/v) FA.

Collected fractions were analyzed by liquid chromatography (LC) -coupled tandem mass spectrometry (MS/MS) using an UltiMate 3000 RSLC nano LC system (Dionex) fitted with a trapping cartridge (µ-Precolumn C18 PepMap 100, 5 µm, 300 µm i.d. × 5 mm, 100 Å) and an analytical column (nanoEase™ M/Z HSS T3 column 75 µm × 250 mm C18, 1.8 µm, 100 Å, Waters). Trapping was carried out with a constant flow of trapping solvent (0.05% trifluoroacetic acid in water) at 30 µL/min onto the trapping column for 6 min. Subsequently, peptides were eluted and separated on the analytical column using a gradient composed of Solvent A ((3% DMSO, 0.1% formic acid in water) and solvent B (3% DMSO, 0.1% formic acid in acetonitrile) with a constant flow of 0.3 µL/min. The outlet of the analytical column was coupled

directly to an Orbitrap Fusion Lumos (Thermo Scientific, SanJose) mass spectrometer using the nanoFlex source.

The peptides were introduced into the Orbitrap Fusion Lumos via a Pico-Tip Emitter 360 µm OD × 20 µm ID; 10 µm tip (CoAnn Technologies) and an applied spray voltage of 2.1 kV, instrument was operated in positive mode. The capillary temperature was set at 275 °C. Only charge states of 4–8 were included. The dynamic exclusion was set to 30 s and the intensity threshold was 5e$^4$. Full mass scans were acquired for a mass range 350–1700 $m/z$ in profile mode in the Orbitrap with resolution of 120,000. The AGC target was set to Standard, and the injection time mode was set to Auto. The instrument was operated in data-dependent acquisition (DDA) mode with a cycle time of 3 s between master scans and MSMS scans were acquired in the Orbitrap with a resolution of 30,000, with a fill time of up to 100 ms and a limitation of 2e5 ions (AGC target). A normalized collision energy of 32 was applied. MS2 data was acquired in profile mode.

### Data analysis
All data were analyzed using the crosslinking module in Mass Spec Studio v2.4.0.3524 (www.msstudio.ca, https://doi.org/10.1074/mcp.O116.058685). Parameters were set as follows: Trypsin (K/R only), charge states 4 − 8, peptide length 7 − 50, percent $E$ value threshold = 50, MS mass tolerance = 10 ppm, MS/MS mass tolerance = 10, elution width = 0.5 min. BS3 cross-links residue pairs were constrained to K on one end and one of KSTY on the other. Identifications were manually validated, and cross-links with an $E$ value corresponding to <0.05% FDR were rejected. The data export from the Studio was filtered to retain only cross-links with a unique pair of peptide sequences and a unique set of potential residue sites. The crosslinks were visualized in 1-D using custom-generated Python scripts (Source Data Fig. 2).

## Yeast two-hybrid assay

The matchmaker GAL4 two-hybrid system 3 (Clonetech Laboratories) was used in this study. The plasmids pGADT7-gtw (also called pVB212) and pGBKT7-gtw (also called pVB213) were used to generate Gal4-Activation domain and Gal4-DNA binding domain fusions, respectively, via Gateway technology.

### Construction of vectors
The HFD fragment from hsCENP-T and bmCENP-T was cloned in fusion with the Gal4-activation domain in the pVB 212 plasmid as follows.

The hsCENP-T-HFD fragment along with the two-helical extension was amplified from cDNA using the primer pairs attb1-hHFD+ext-F/ attb1-hHFD+ext-R. Similarly, the bmCENP-T-HFD fragment was amplified from pIZV5-BomCENP-T-3xFLAG plasmid (Cortes-Silva et al, 2020) using the primer pairs attb1-BmHFD+ext-F/ attb1-BmHFD+ext-R. The purified PCR products were directly cloned in frame with the Gal4-activation domain in the destination vector pVB212 using the "one tube" protocol described in the Gateway technology user guide (Invitrogen). The reaction mixture was used to transform competent DH5α cells, and the transformants were selected in LB plates supplemented with 100 µg/mL ampicillin. Clones were verified for proper integration by PCR and Sanger sequencing. The resulting plasmids were named pVB212-hsT and pVB212-bmT, respectively.

The hsCENP-W and scCENP-W ORFs were cloned in frame with Gal4-DNA binding domain in the pVB213 plasmid as follows.

The hsCENP-W ORF was amplified from pFastBac_hCenpW-StrepTag using the primer pairs attb1-hCenpW-F/attb1-hCenpW-R. Similarly, the scCENPW ORF was amplified from pSW108 (Schleiffer et al, 2012) using the primer pairs attb1-ScCenpW-F/attb1-ScCenpW-R. The purified PCR products were directly cloned in frame with the Gal4-DNA binding domain in the destination vector pVB213 using the "one tube" protocol described in the Gateway technology user guide (Invitrogen). The reaction mixture was used to transform competent DH5α cells, and the transformants were selected in LB plates supplemented with 30 μg/mL Kanamycin. Clones were verified for proper integration by PCR and Sanger sequencing. The resulting plasmids were named pVB213-hsW and pVB213-scW, respectively.

### Transformation and spot dilution assay

The transformation of yeast strain AH109 was performed using the protocol provided in the Matchmaker GAL4 two-hybrid system 3 (Clontech) manual. The transformants were selected in SD-leu-trp plates after incubation at 30 °C for 48 h. For the spot dilution assay, the indicated strains were propagated in Sd-leu-trp broth overnight at 30 °C. Serial dilutions were made based on the $OD_{600}$ such that the desired number of cells could be spotted in a 5 μL volume. Spotting was done on SD-leu-trp (loading control), SD-leu-trp-his (low stringency), and SD-leu-trp-his-ade (high stringency) dropout plates. The plates were photographed after incubation at 30 °C for 48 h.

## Lepidopteran cell lines and culture conditions

Cultured silkworm ovary-derived BmN4-SID1 cell lines (RRID:CVCL_Z091) (Kobayashi et al, 2012) were maintained in Sf-900 II SFM medium (Gibco Cat#10902-088) supplemented with 10% fetal bovine serum (Eurobio Cat#CVFSVF0001), antibiotic-antimycotic (Gibco Cat#15240-062), and 2 mM L-glutamine (Gibco Cat#25030-024) at 27 °C. Sf9 cells (Gibco Cat#12659017) harboring a LacO array (Sf9-LacO) (Cortes-Silva et al, 2020) were maintained in Sf-900 II SFM medium (Gibco Cat#10902-088) supplemented with antibiotic-antimycotic (Gibco Cat#15240-062) and 2 mM L-glutamine (Gibco Cat#25030-024) at 27 °C.

## Plasmid constructs

### Constructs to express 3xFLAG-recoded bmCENP-T^WT, bmCENP-T^patch1, and bmCENP-T^Δtail in BmN4 cells

The pHyg_1kbTubulinprom_eGFP used as the base plasmid for cloning different versions of bmCENP-T was constructed as follows. Each of these clones, the Gibson assembly reaction mixture was used to transform competent DH5α cells, and were selected on LB plates supplemented with Hygromycin (200 μg/mL). Correct integrants were verified by sequencing.

*pHyg-GFP*: The hygromycin gene (1055 bp) was amplified from the pIRESHyg3 plasmid using the primers GC129-F/GC130-R and cloned into the pIVZ5_eGFP plasmid by Gibson assembly. The vector fragment (3191 bp) for this reaction was amplified using the primers GC127-F/GC128-R.

*pHyg_Tubulinprom-eGFP*: The tubulin promoter (4957 bp upstream of Tubulin ORF) was then cloned into the pHyg-GFP

plasmid by Gibson assembly. For this reaction, the insert (4957 bp) was amplified from cDNA using primers Tubulin_Prom_Fcloning/Tubulin_Prom_Rcloning, and the vector (pHyg_GFP, 3661 bp) was amplified using pIVZ5_Hyg_GFP_F/ pIVZ5_Hyg_GFP_R.

*pHyg_1 kbTubulinprom_eGFP*: The promoter in the above plasmid was further optimized to contain only the region 1kb upstream of the Tubulin ORF. The 3957 bp sequence upstream of this 1 kb region in the pHyg_Tubulinprom-eGFP plasmid were eliminated by a Gibson reaction with a single fragment (4682 bp) amplified from this vector using the primers GC161-F/GC162-R.

The endogenous CENP-T sequence was amplified and cloned downstream of the tubulin promoter to replace eGFP by Gibson assembly. The insert with a 3xFLAG tag at the N-terminus was amplified from the template pHyg_CENP-Tprom_3xFLAG-CENP-T using the primers PL075-F/PL078-R (3157 bp), and the vector was amplified from the plasmid pHyg_1kbTubulinprom_eGFP with primers GC104-F/PL008-R (3942 bp). The resulting plasmid was named pHyg_1kbTubulinprom_3xFLAG-CENP-T.

The endogenous CENP-T from the above plasmid was then replaced by a recoded RNAi-resistant version as follows. The recoded bmCENP-T^WT was amplified from pIVZ5-recodedCENP-T-3xFLAG (Cortes-Silva et al, 2020) using VA129-F/GC244-R (3091 bp). The vector pHyg_1kbTubulinprom_3xFLAG-CENP-T was amplified using VA136-F/PL022-R (4030 bp) such that the endogenous CENP-T was excluded. A Gibson reaction with these two fragments resulted in the plasmid pHyg_1kbTubulinprom_3xFLAG-recodedCENP-T^WT.

Similarly, the recoded bmCENP-T^patch1 sequence was amplified from an intermediate vector (described below) pHyg_recodedCENP-T^patch1_untagged using the primers VA129-F/GC244-R (3091 bp) to replace the endogenous CENP-T to obtain the plasmid pHyg_1kbTubulinprom_3xFLAG-recodedCENP-T^patch1.

The intermediate vector pHyg_recodedCENP-T^patch1_untagged was generated as follows. The first 1154 bp of recoded CENP-T^WT was amplified from the plasmid pIVZ5-recodedCENP-T-3xFLAG using primers PL 098 F/ SR142. The remaining 1902 bp of the CENP-T gene was synthesized to contain the desired mutations. The synthesized DNA was amplified using primers SR140/ SR144. The two fragments were then cloned in frame into the pHyg-GFP vector by Gibson assembly, for which the vector was amplified with PL111-F/PL008-R (3493 bp).

The plasmid to express 3xFLAG-tagged bmCENPT-T^Δtail was constructed as follows. The plasmid pHyg_1kbTubulinprom_3xFLAG-recodedCENP-T^WT was used as a template to amplify the tail-deleted bmCENP-T ORF using the primers PL075-F/SR156-3 and cloned as a *Bam*HI-*Xho*I fragment into the same plasmid to replace recoded bmCENP-T^WT with bmCENP-T^Δtail. The loss of restriction site *Sph*I from the tail was used to screen positive transformants. The clones were further verified by sequencing.

### Constructs to express sfCENP-T^Δtail-3xFLAG in Sf9 cells

A previously reported plasmid pIBV5_SfCENP-T^WT-3xFLAG vector (Cortes-Silva et al, 2020) was used as a control in our study. Using the primers GC238-F/GC240-R, this plasmid was amplified such that the sequence encoding the predicted CENPK-interacting part in sfCENP-T (aa 1288-STOP) was excluded. The 7355 bp amplicon was gel-purified, phosphorylated, and ligated using T4 DNA ligase. Transformants were selected on LB plates

supplemented with 100 μg/mL ampicillin. The resulting plasmid pIBV5_sfCENP-T$^{\Delta tail}$-3xFLAG was verified by sequencing.

## Transfections and RNAi-mediated knockdown

BmN4-SID1 cells were grown on coverslips and transfected using 1–2 μg of plasmid DNA using XtremeGene (Roche). After 2 days, 400 pg/μl dsRNA targeting the endogenous *B. mori* CENP-T was added to the media (Cortes-Silva et al, 2020). Cells were split 1:2 after another 3 days and grown for 2 days before fixation. Similarly, Sf9-LacO cells were grown on coverslips and co-transfected using 1–2 μg of plasmid DNA (LacI-GFP-sfCENP-I and sfCENP-T-3xFlag or sfCENP-T$^{\Delta tail}$-3xFlag) using XtremeGene (Roche) and grown for 5 days.

## Immunofluorescence

Cells were grown on glass coverslips and fixed 4% PFA, followed by permeabilization using 0.3% Triton X-100 in PBS and blocked in 3% BSA-PBS. The following antibodies were used: rabbit polyclonal anti-CENP-T (Cortes-Silva et al, 2020) at a dilution of 1:1000, anti-FLAG M2 mouse monoclonal (Sigma Cat#F1804; RRID:AB_262044) at 1:1000, anti-phospho Histone H3-Ser10 rat monoclonal (Sigma Cat#MABE939) at 1:1000. For fluorescent conjugated secondary antibodies, we used goat anti-rat IgG Alexa Fluor 488 (Thermo Fisher Scientific Cat#A-11006; RRID:AB_2534074) at 1:1000, goat anti-mouse IgG Alexa Fluor 568 (Thermo Fisher Scientific Cat#A-11004; RRID:AB_2534072) at 1:1000, goat anti-rabbit IgG Alexa Fluor 633 (Thermo Fisher Scientific Cat#A-21070; RRID:AAB_2535731). DNA was stained with DAPI (Sigma Cat#D9542), and samples were mounted in Vectashield Antifade Mounting Medium (Vector Laboratories Cat# H-1000; RRID:AB_2336789).

## Microscopy- image acquisition and analysis

Images to quantify mitotic index upon CENP-T depletion (Fig. EV4A) in untransfected cells, and those expressing recoded 3xFLAG-bmCENP-T$^{WT}$ or 3xFLAG-bmCENP-T$^{patch1}$ were acquired on a Zeiss Axiovert Z1 light microscope. Z-sections were acquired at 0.2 μm steps using 40× oil objective. Images to characterize the mitotic defects (Fig. EV4B) in these slides were acquired using 100×1.4 NA oil objective with 0.2 μm Z steps. Representative images for Fig. EV4C were taken using a Leica Thunder microscope with a 63x objective (oil immersion, NA 1.4).

To quantify mitotic index, all the cells in a given field were classified into untransfected interphase cells (DAPI-stained nuclei, α−phosphoH3-S10P negative, α−FLAG negative), untransfected mitotic cells (α−phosphoH3-S10P positive, α−FLAG negative), transfected interphase cells (α−phosphoH3-S10P negative, α−FLAG-positive), and transfected mitotic cells (α−phosphoH3-S10P positive, α−FLAG-positive). The percentage of mitotic cells in the untransfected and transfected populations were then calculated and plotted using GraphPad Prism. The values were derived by counting ~1000 cells.

To estimate the frequency of defective mitosis, untransfected mitotic cells (α-phosphoH3-S10P positive) or transfected mitotic cells (α-phosphoH3-S10P and α-FLAG-positive) were classified into three categories—with no apparent mitotic defects, with

alignment defects, or having congression defects (representative images in EV4C). The percentage of these defects was calculated and plotted using GraphPad Prism.

To quantify the fluorescent signal intensities for LacO/LacI tethering assays in Sf9-LacO cells images of transfected cells both expressing LacI-GFP-sfCENP-I and the respective sfCENP-T-3xFlag wildtype or truncated constructs were acquired in a Leica Thunder microscope using a 63x objective (oil immersion, NA 1.4). Briefly, the mean fluorescence intensities of sfCENP-T-3xFlag or sfCENP-T$^{\Delta tail}$-3xFlag signals were first measured in the region overlapping LacI-GFP-sfCENP-I foci (visualized by the GFP signals). Then, the background mean fluorescence intensity of CENP-T was determined as the average in three random circular regions of fixed size (10 × 10 pixels) placed in the nuclear area. Both the mean fluorescence intensities overlapping the LacI foci or the endogenous loci were corrected with this background value. For statistical analysis, a Student's *t* test (unpaired, unequal variance) was used. Differences were considered statistically significant at values of *P* values < 0.05.

## Identification of CENP-T and CENP-W homologs in Arthropods

Annotated CENP-T sequences from two acariformes, *Rhipicephalus zambeziensis* (A0A224Y4D1) and *Ornithodoros erraticus* (A0A293LVR6) were available in the uniport database. To identify homologs of CENP-T across arthropods about 203 amino acids from the C-terminus of *R. zambeziensis* CENP-T was used as a query in a DELTA blast search with a taxonomic filter that restricted the search to Arthropoda, but excluded the hits from Hexapoda. Hits from related Acariformes like *Rhipicephalus macroplus* (XP_037286617.1), *Dermacentor silvarum* (XP_049524110.1), and *Dermacentor andersoni* (XP_054925727.1) were picked up and confirmed as CENP-T homologs in reciprocal searches. Besides Acariformes, we could also detect an annotated CENP-T homolog from *Limulus polyphemus* (horseshoe crabs, XP_022239208.1), *Penaeus vannamei* (shrimps, XP_027236760.1), and *Helicorthomorpha holstii* (millipede, Hho_012284-T1). Further iterations resulted in hits that were predominantly annotated as Histone H4-like proteins.

To identify CENP-W homologs, a similar DELTA BLAST search with the taxonomic filter described as above was performed using the hsCENPW protein sequence (Q5EE01). This search resulted in only two hits, one of which was CENP-W-like protein from *Limulus polyphemus* (XP_022257414.1). The other hit was annotated as Protein Dr1 from a mite *Galendromus occidentali* and failed to fetch CENP-W in reciprocal searches. We then used the CENP-W hit from *L. polyphemus* in a HMMER search (https://toolkit.tuebingen.mpg.de/tools/hmmer) (Gabler et al, 2020) to identify homologs from *P. vannamei* (XP_027215827.1) and *H. holstii* (Hho_014734-T1). In neither of these searches, we could find a CENP-W homolog in the acariformes from which the CENP-T sequences described above were detected.

## AlphaFold predictions and the protein sequences for which models were generated

The Colab notebook version of AlphaFold2 was used to generate all the models reported in this study (colab.research.google.com)

(Jumper et al, 2021). The top-ranked model for each protein sequence query is reported in the manuscript. The models were visualized using ChimeraX 1.4 (Pettersen et al, 2021).

Alphafold 3 webserver (Abramson et al, 2024) was used to predict (i) the structure of bmCENP-T homodimer, and (ii) the interaction between CENP-T-C' and CENP-K [using the sequences of CENP-T-C' (aa 889–1015) and the CENP-HIK head from *B. mori*]. The corresponding sequences in sfCENP-T were identified by structural superimposition using ChimeraX 1.4.

For the illustrations depicting the hydrophobic core in bmCENP-T-HFD monomer and the ggCENP-TW dimer in Fig. 2, the side chains of only the hydrophobic residues were displayed to highlight the proximity of the histone fold extension and the histone-fold domain in the case of bmCENP-T, and between CENP-W and the surrounding helices of CENP-T in the ggCENP-TW heterodimer.

All protein sequence ID of CENP-T and CENP-W homologs from Figs. 4 and EV5 are in Dataset EV2. The CENP-T sequences used for Fig. EV2F are also in Dataset EV2.

## Shannon's entropy analyses

To calculate Shannon's diversity index over the C-termini of vertebrate and insect CENP-T proteins we aligned 330 vertebrate CENP-T obtained from Interpro (Paysan-Lafosse et al, 2023) searching proteins with domain architectures corresponding to Q96BT3 and 117 lepidopteran CENP-T from species sequenced as part of the Darwin Tree of Life project (Blaxter, 2022), 226 hymenopteran CENP-T proteins obtained from the Hymenoptera Genome Base (Walsh et al, 2022) and Ensemble Rapid Release as well as several CENP-T proteins from additional insect species identified in (Cortes-Silva et al, 2020). The alignments were trimmed to only include the C-termini, and all positions that had gaps were removed. Redundant sequences were removed afterwards. For each alignment (vertebrate and insect) Shannon's diversity index was calculated using the protein Variability Server (http://imed.med.ucm.es/Tools/pvs.html). For comparison between the two sets of sequences, Shannon's diversity indices for each position were divided by the median of all indices for each set. Full-length sequences for the vertebrate and insect CENP-T proteins are available in the Dataset EV2.

## Immunoblotting

### For solubility testing of E. coli cells

Five OD cells were harvested, washed, and resuspended in 1 mL of 1×PBS supplemented with 5 mM $MgCl_2$ and DNAse. The suspension was then sonicated using a thin probe with the following settings. Amplitude – 25%, Pulse: 2 s ON/OFF, Time: 3×1 min cycle with chilling the tubes on ice for 5 min in between cycles. The sample was then centrifuged at $20,000 \times g$ for 20 min at 4 °C. The supernatant was saved, and the pellet was resuspended in 0.5 mL of lysis buffer (0.1 N NaOH, 1% SDS) and mixed thoroughly by pipetting. For immunoblot, 20 μL of soluble fraction and 10 μL of resuspended pellet fraction were used. Mouse anti-HIS antibodies (Abcam) and Goat anti-mouse 800CW (LiCOR) antibodies were used at dilutions 1:2000 and 1:10,000, respectively, as primary and secondary antibodies. The blot was developed using the ChemiDoc imaging system (Bio-Rad).

### For testing the expression of fusion proteins in Y2H strains

Three OD equivalent cells were harvested and resuspended in 14% TCA. The samples were frozen in −20 °C overnight. The samples were thawed on ice and then centrifuged at 10,000 rpm at 4 °C for 10 min. The pellets were washed with 80% ice-cold acetone twice, and then the pellet was left to air-dry. The pellet was resuspended in 40 μL lysis buffer (0.5 N NaOH, 1% SDS) by intermittent vortexing. The samples were boiled after adding 10 μL of 5× SDS-loading dye. Goat anti-myc and Rabbit anti-HA primary antibodies were each used at 1: 1500 dilution. The secondary antibodies were used at 1:10,000 dilution.

## Data availability

The mass spectrometry proteomics data have been deposited to the ProteomeXchange Consortium via the PRIDE partner repository with the dataset identifier PXD063732.

The source data of this paper are collected in the following database record: biostudies:S-SCDT-10_1038-S44319-025-00603-5.

## Peer review information

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

## Acknowledgements

SRS acknowledges the financial support from EMBO (LTF 505-2021). IAD receives salary support from the CNRS. Research in the Drinnenberg lab is supported by the Labex DEEP ANR-11-LABX-0044 part of the IDEX Idex PSL ANR-10-IDEX-0001-02 PSL, an ATIP-AVENIR Research grant, the Fondation pour la Recherche Médicale (FRM) and Fondation Schlumberger (FSER202202015420), Institut Curie, the ERC (CENEVO-758757). Research in Sekulić lab is supported by the NCMM and the Research Council of Norway (grant numbers 187615 and 325528). We thank Gaétan Cornilleau from the Drinnenberg lab for generating the plasmids for the yeast two-hybrid experiment. We thank the EMBL proteomics platform for support with CLMS analysis, Ahmad El Marjou from the Curie recombinant protein platform, Carlos Kikuti (Institut Curie, Paris), and Alexandre Pozze (IBPC, Paris) for help with the SEC-MALS analyses, and Vikram Alva-Kullanja (Max Planck Institute for Biology, Tubingen) for advice on the evolution of the histone-fold domain. We also acknowledge the support from Fabien Ferrage, Guillaume Bouvignies, and Theodore Bellon (Ecole Normale Supérieure - Département de Chimie, Paris) for help with preliminary NMR analysis. We thank Stanislau Yatskevitch (Genentech, USA) for critical reading of the manuscript. We would also like to thank Life Science Editors for editing services (www.lifescienceeditors.com).

## Author contributions

**Sundar Ram Sankaranarayanan**: Conceptualization; Data curation; Formal analysis; Investigation; Visualization; Methodology; Writing—original draft; Writing—review and editing. **Jonathan Ulmer**: Formal analysis; Investigation; Methodology. **Anna Mørch**: Formal analysis; Investigation; Methodology. **Ahmad Ali-Ahmad**: Formal analysis; Investigation; Visualization; Methodology; Writing—review and editing. **Nikolina Sekulić**: Conceptualization; Resources; Supervision; Funding acquisition; Methodology; Project administration; Writing—review and editing. **Ines Anna Drinnenberg**: Conceptualization; Resources; Data curation; Supervision; Funding acquisition; Visualization; Methodology; Writing—original draft; Project administration; Writing—review and editing.

Source data underlying figure panels in this paper may have individual authorship assigned. Where available, figure panel/source data authorship is listed in the following database record: biostudies:S-SCDT-10_1038-S44319-025-00603-5.

## Disclosure and competing interests statement

The authors declare no competing interests.

# Expanded View Figures

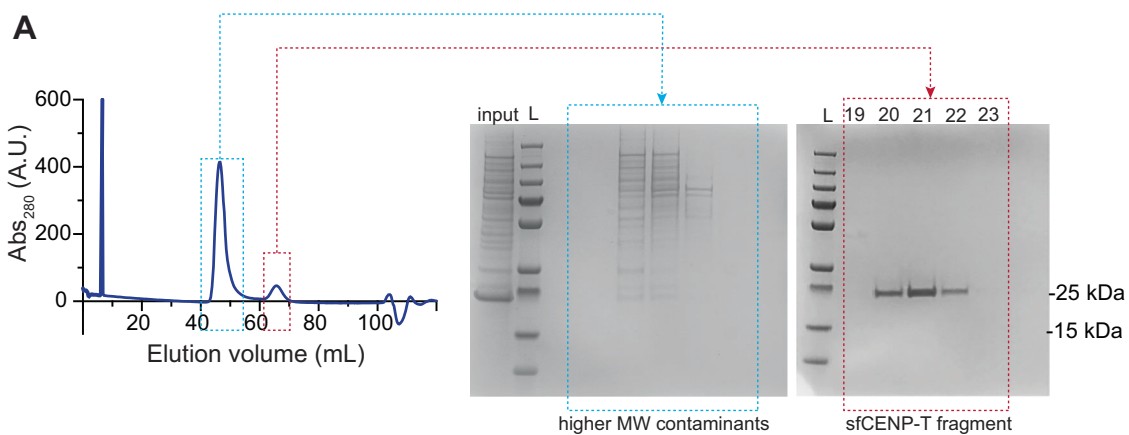

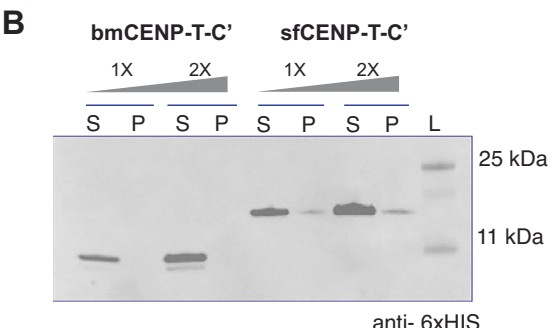

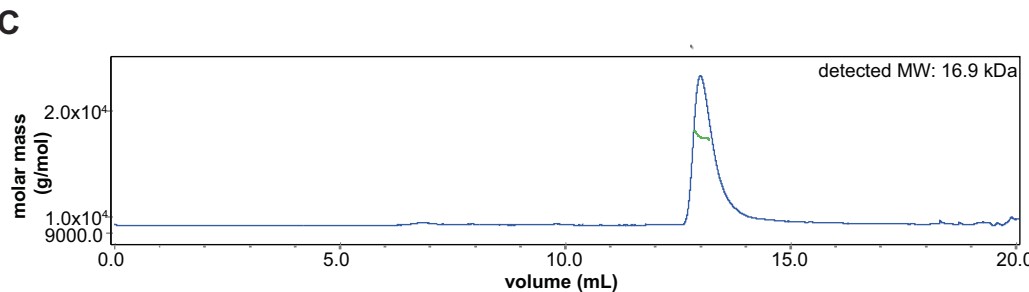

**Figure EV1.  Solubility of lepidopteran histone fold in SF9 and *E. coli* cells.**

(A) An expanded view SEC profile and the corresponding SDS-PAGE analysis presented in Fig. 1C to highlight all the indicated fractions of 6xHis-sfCENP-T$^{1147-1314}$ purification. The SDS page profile of the peak in red box from Fig. 1C is reused in this panel to compare with the contaminating proteins eluted in the peak boxed in blue. Input: a fraction of the sample loaded into the SEC column, L: molecular weight marker. (B) Western blot analysis of the soluble (S) and pellet (P) fractions of *E. coli* BL21 cells expressing 6xHis-bmCENP-T (16.2 kDa) and 6xHis-sfCENP-T (21.5 kDa) fragment with anti-His antibodies. L: molecular weight marker. (C) SEC-MALS analysis of bmCENP-T$^{894-1016}$ fragment reveals the monomeric state of the protein. The molecular weight inferred from MALS (16.9 kDa) is represented as a blue line overlaid on the SEC peak for this protein. Source data are available online for this figure.

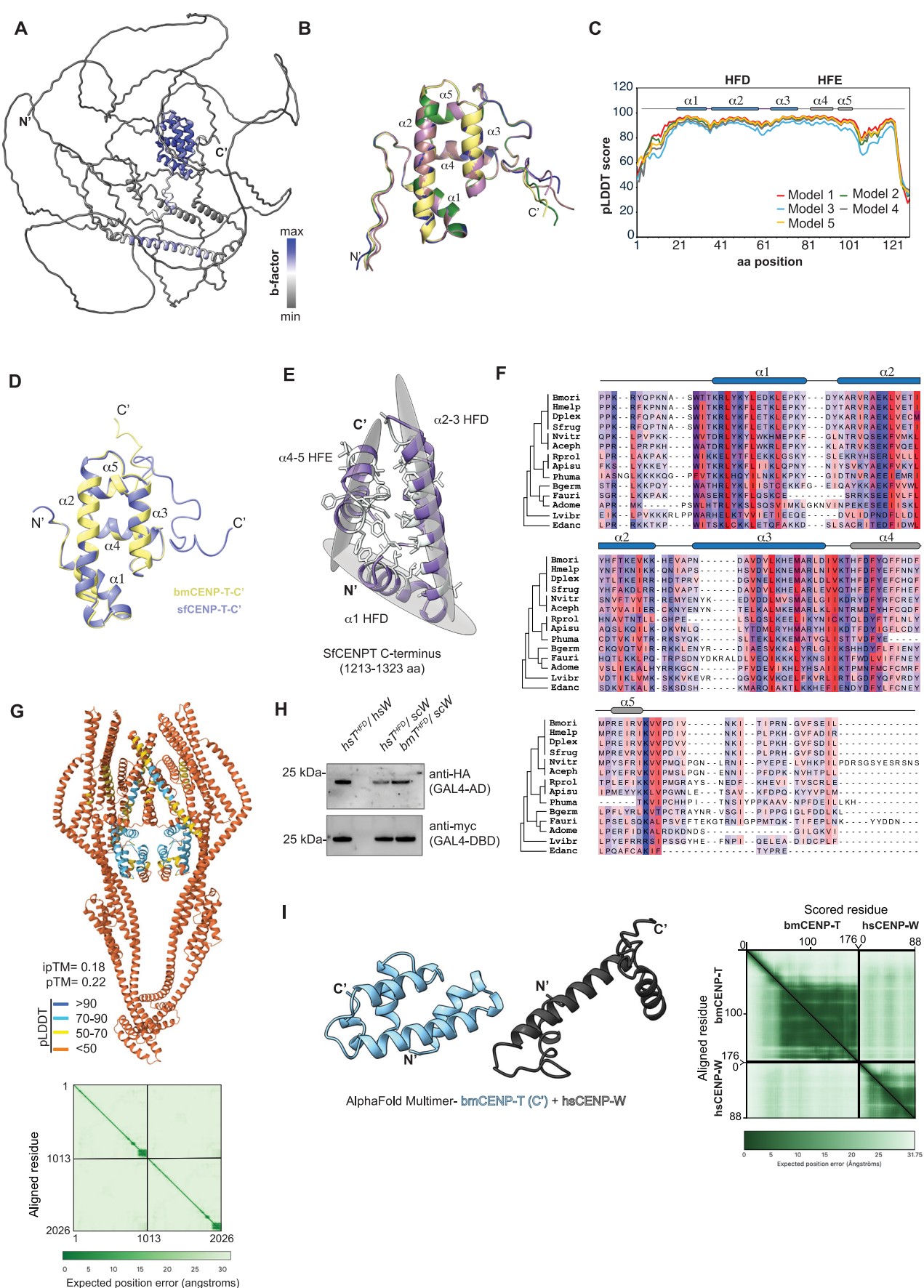

**Figure EV2.  Structural basis for the stability of bmCENP-T in the absence of an interacting partner.**

(**A**) AlphaFold predicted structure of full-length bmCENP-T colored based on the confidence scores for each residue (bfactor-pLDDT score). (**B**) Overlay of the top five bmCENP-T models generated by AlphaFold. The helices of the HFD (α1–3) and the extension (α4-5) are indicated. The sequences upstream of the HFD in these models are hidden for clarity. (**C**) The pLDDT plot for the five models shown in (**A**). The regions corresponding to the helices of the HFD, and the extension are highlighted by a line diagram above the plot. (**D**) The best-ranked AlphaFold models for bmCENP-T and sfCENP-T are overlaid to highlight structural similarities at the C-terminus. Only the regions corresponding to the HFD and the HFE are shown in the panel. (**E**) Similarity in the stabilization of the monomeric HFD in sfCENP-T by the hydrophobic residues in the extension as shown for bmCENP-T in Fig. 2C. (**F**) Multiple sequence alignment of C-terminal fragment of CENP-T containing the HFD and HFE from several insect orders. The alignment was generated by MAFFT and was colored using the hydrophobicity scheme in JalView. The line diagram above the sequences indicates the helices of HFD (α1–3) and HFE (α4-5) as in *B. mori*. The cladogram represents insect orders where CENP-T has been detected. The sequences species and the orders they represent are as follows: Lepidoptera (*B. mori, Heliconius melpomene, Danaus plexippus, S. frugiperda*), Hymenoptera (*Nasonia vitripennis, Atta cephalotes*), Hemiptera (*Rhodnius prolixus, Acyrthosiphon pisum*), Phthiraptera (*Pediculus humanus corporis*), Blattoeda (*Blatella germanica*), Dermaptera (*Forficula auricularia*), Orthoptera (*Acheta domestica*), Odonata (*Libuellua vibrans*), Ephemeroptera (*Ephemera dancia*). (**G**) The AlphaFold3 model of a homodimeric full-length bmCENP-T colored based on pLDDT scores. The pTM and ipTM values generated by the AlphaFold webserver are indicated. The predicted aligned error matrix is presented below the model. It should be noted that, unlike the prediction for bmCENP-T monomer, the N-terminus of CENP-T is predominantly helical in the homodimer, albeit with a low complex confidence score. (**H**) Whole cell extracts from cells used in the spot dilution assay shown in Fig. 2G were analyzed by immunoblotting using anti-Myc and anti-HA antibodies. (**I**) The inability of a monomeric HFD (from bmCENP-T) to interact with a canonical HFD (from hsCENP-W) as revealed by AlphaFold multimer. The predicted aligned error (PAE) matrix is presented on the right. Source data are available online for this figure.

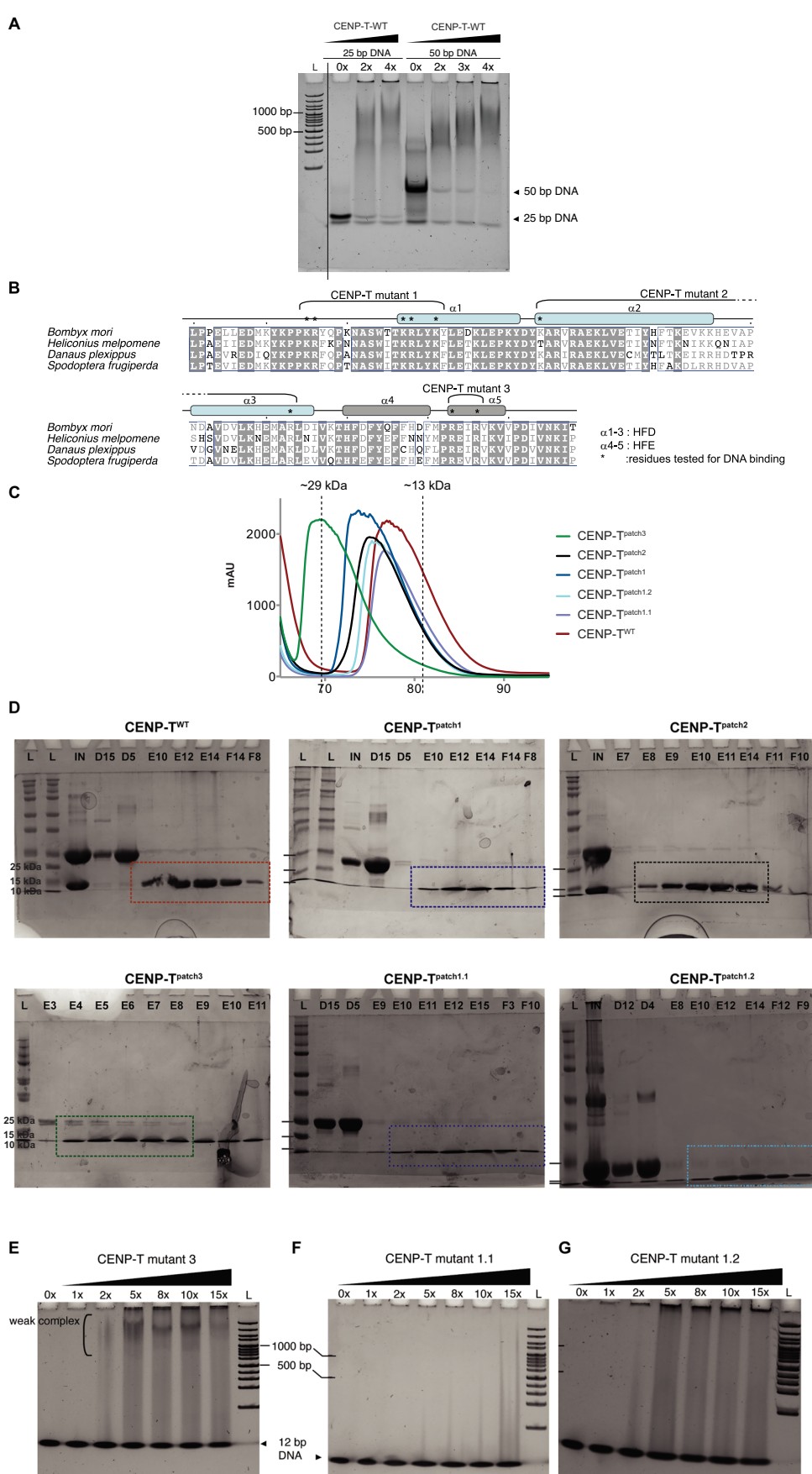

◀ **Figure EV3.  Residues in patch 2 and patch 3 are dispensable for DNA binding.**

(A) Native PAGE gels of DNA titration using different molar ratios of bmCENP-T$^{894-1016}$-WT and a 25 bp or a 50 bp DNA fragment at 100 mM NaCl. Lane L: 100 bp DNA marker. (B) Multiple sequence alignment of the C' fragment of CENP-T between 4 lepidopteran species. Identical residues are highlighted with a gray background. The residues tested for DNA binding function are marked with an asterisk. (C) The SEC elution profile of the WT and mutant versions of bmCENP-T used in the DNA binding assays in Figs. 3 and  EV3. (D) The Coomassie stained SDS-polyacrylamide gels containing the SEC fractions for the purification of the indicated versions of bmCENP-T after cleaving the GST tag. The quality of each sample studied in S3C (colored boxes) is visually represented. L- Ladder, IN- Input sample, alphanumeric labels starting with D/E/F are different fractions collected in the experiment. (E–G) Native PAGE gels of DNA titration using different molar ratios of bm CENP-T$^{894-1016}$ -patch 3 mutant (C), bmCENP-T$^{894-1016}$ patch 1.1 mutant with the substitutions K895A, R896A, and K907A (D) and bmCENP-T$^{894-1016}$ patch 1.2 mutant with the substitutions R908S and K911A (E). Lane L: 100 bp DNA marker.

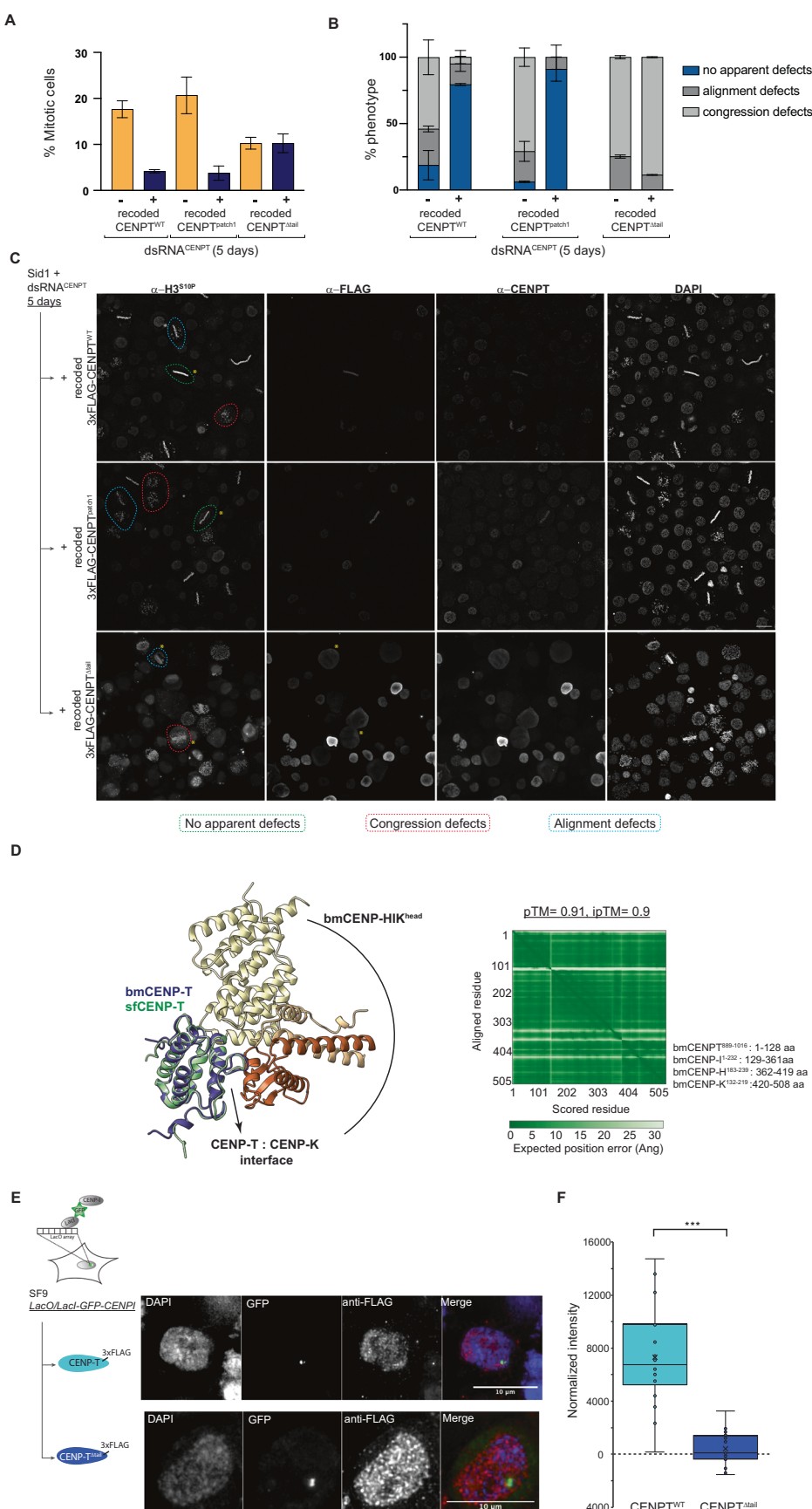

◀ **Figure EV4. Properties and roles of the *B. mori* CENP-T as part of the kinetochore in *B. mori* cells.**

(A) Mitotic index of untransfected (gray) or transfected (blue) cells expressing RNAi-resistant versions of bmCENP-T$^{WT}$, bmCENP-T$^{patch1}$, or bmCENP-T$^{\Delta tail}$ upon depletion of endogenous CENP-T. (B) Fraction of cells with or without mitotic defects, and the nature of defects observed (congression or alignment defects) upon endogenous CENP-T depletion in lines expressing bmCENP-T$^{WT}$, bmCENP-T$^{patch1}$, or bmCENP-T$^{\Delta tail}$. (C) Representative examples of untransfected or transfected cells expressing bmCENP-T$^{WT}$, bmCENP-T$^{patch1}$, or bmCENP-T$^{\Delta tail}$ in mitosis stained against phospho histone H3 (anti-H3$^{S10P}$), anti-FLAG, anti-CENP-T, and DAPI. Scale bar, 10 μm. (D) AlphaFold multimer model of the interaction between bmCENP-T$^{889-1016}$ and the bmCENP-T-HIK$^{head}$ constituted by bmCENP-T-I$^{1-232}$, bmCENP-T-H$^{183-239}$, and bmCENP-T-K$^{132-219}$. The confidence scores and the predicted aligned error matrix are presented on the right. The AlphaFold model of full-length sfCENP-T was superimposed over the CENP-HIK$^{head}$:T model. The disordered N terminus of sfCENP-T upstream of the histone fold were hidden for clarity. (E) Representative images of DAPI-stained cells showing localization pattern of 3xFLAG-tagged sfCENP-T$^{WT}$ (top) or sfCENP-T$^{\Delta tail}$ (bottom) in Sf9-LacO lines transiently expressing LacI-GFP-sfCENPI constructs. Merge: DNA (blue), expressing LacI-GFP-sfCENPI (green) and 3xFLAG-tagged sfCENP-T$^{WT}$ (top) or sfCENP-T$^{\Delta tail}$ (red). Scale bar: 10 μm. (F) Quantifications of normalized mean fluorescence intensity of 3xFLAG-tagged sfCENP-T$^{WT}$ or sfCENP-T$^{\Delta tail}$ at the LacI foci. Statistical significance was tested using the Student's *t* test (*** indicates $P = 3.8 \times 10^{-7}$). The edges of the box mark the 25th and 75th percentile and the line inside marks the median value. Source data are available online for this figure.

**A**

Vertebrates

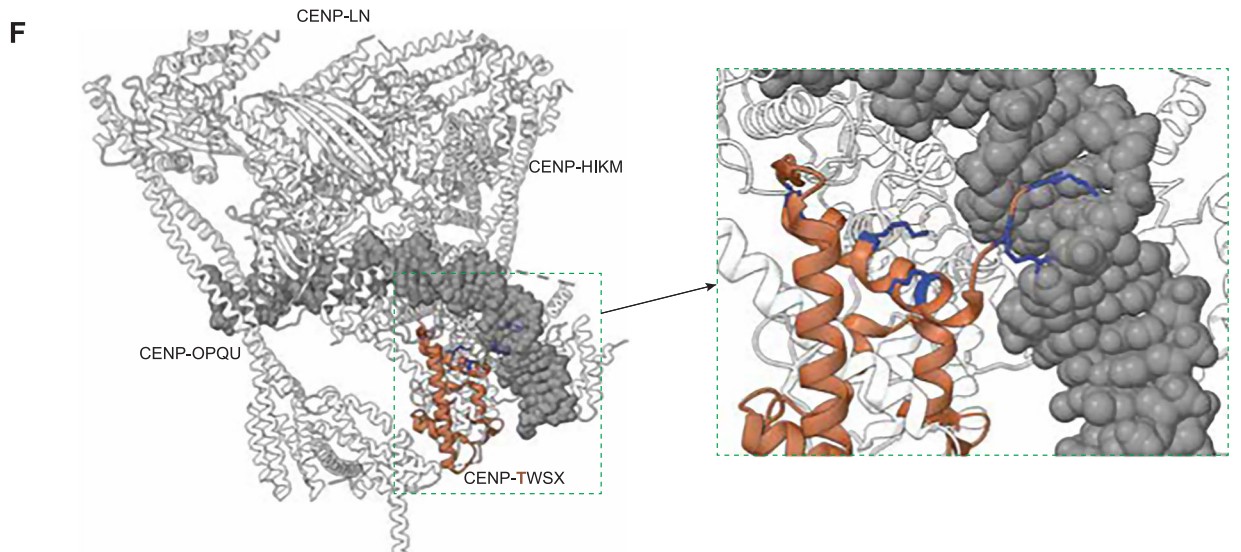

**B**

Insects

**C**

*Rhipicephalus macroplus*

**D**

*Dermacentor silvarum*

**E**

*Ornithodoros erraticus*

**F**

CENP-LN

CENP-HIKM

CENP-OPQU

CENP-TWSX

**Figure EV5. Diversity in the sequence and structure of the monomeric HFD in Arthropoda.**

(A, B) Comparative analysis of sequence diversity in vertebrate and insect CENP-T using Shannon's entropy analysis. The gray dots represent the normalized Shannon's diversity ratio for each amino acid in the C' of CENP-T. The trendline represents the moving average of 5 datapoints. (C–E) The AlphaFold model for CENP-T from three acariformes, in which CENP-W remains undetected. The canonical HFD structure in each of the three species is highlighted in beige, pink, and green, respectively. The HFE is colored gray and a part of the N' of the protein that blocks the interaction space for CENP-W is shaded blue. For protein IDs see Dataset EV2. (F) The structure of human CCAN reported in Yatskevitch et al (2022) highlighting the proximity of α1 of CENP-T HFD (CENP-T is colored brown) to DNA (gray). The side chains of the positively charged amino acids in this region are shown in blue. The illustration was adapted from PDB 7R5S.

