## [Peer Review File · EMBO Reports]

Insects evolved a monomeric histone-fold domain in the CENP-T protein family

Sundar Ram Sankaranarayanan, Jonathan Ulmer, Anna Mørch, Ahmad Ali-Ahmad, Nikolina Sekulic, and Ines Drinnenberg

Corresponding author(s): Ines Drinnenberg (ines.drinnenberg@curie.fr)

Review Timeline:

Submission Date:	14th Nov 24
Editorial Decision:	18th Dec 24
Revision Received:	1st May 25
Editorial Decision:	28th May 25
Revision Received:	15th Sep 25
Accepted:	8th Oct 25

Editor: Yehu Moran

Transaction Report:

Dear Dr. Drinnenberg

Thank you for the submission of your manuscript to EMBO Reports. We have now received the full set of referee reports as well as referee cross-comments that are all pasted below.

As you will see, the referees acknowledge that the findings are potentially interesting. However, they do raise a series of significant comments that require your attention.

I would thus like to invite you to revise your manuscript with the understanding that the referee concerns must be fully addressed and their suggestions taken on board. Please address all referee concerns in a complete point-by-point response. Acceptance of the manuscript will depend on a positive outcome of a second round of review (preferably by the same referees). It is EMBO Reports policy to allow a single round of major revision only and acceptance or rejection of the manuscript will therefore depend on the completeness of your responses included in the next, final version of the manuscript.

We realize that it is difficult to revise to a specific deadline. In the interest of protecting the conceptual advance provided by the work, we recommend a revision within 3 months (20th Mar 2025). Please discuss the revision progress ahead of this time with the editor if you require more time to complete the revisions.

- 1) A data availability section providing access to data deposited in public databases is missing. If you have not deposited any data, please add a sentence to the data availability section that explains that.
- 2) Your manuscript contains statistics and error bars based on $n=2$. Please use scatter blots in these cases. No statistics should be calculated if $n=2$.

<<https://www.embopress.org/page/journal/14693178/authorguide#expandedview>>

5) a complete author checklist, which you can download from our author guidelines

<<https://www.embopress.org/page/journal/14693178/authorguide>>. Please insert information in the checklist that is also reflected in the manuscript. The completed author checklist will also be part of the RPF.

6) Please note that all corresponding authors are required to supply an ORCID ID for their name upon submission of a revised manuscript (<<https://orcid.org/>>). Please find instructions on how to link your ORCID ID to your account in our manuscript tracking system in our Author guidelines

<<https://www.embopress.org/page/journal/14693178/authorguide#authorshipguidelines>>

10) Regarding data quantification (see Figure Legends:

<https://www.embopress.org/page/journal/14693178/authorguide#figureformat>)

12) All Materials and Methods need to be described in the main text using our 'Structured Methods' format, which is required for all research articles. According to this format, the Methods section includes a Reagents and Tools Table (listing key reagents, experimental models, software and relevant equipment and including their sources and relevant identifiers) followed by a Methods and Protocols section describing the methods using a step-by-step protocol format. The aim is to facilitate adoption of the methodologies across labs. More information on how to adhere to this format as well as a downloadable template (.docx) for the Reagents and Tools Table can be found in our author guidelines:

An example of a Method paper with Structured Methods can be found here: <https://www.embopress.org/doi/full/10.1038/s44320-024-00037-6#sec-4>

We would also welcome the submission of cover suggestions, or motifs to be used by our Graphics Illustrator in designing a

cover.

I look forward to seeing a revised form of your manuscript when it is ready.

Yours sincerely,

Yehu Moran
Academic Editor
EMBO Reports

Referee #1:

Sankaranarayanan et al present a nicely focused exploration of altered properties of the CENP-T protein in insects, with a focus on Lepidopteran model species.

Along with CENP-A and CENP-C, insects have lost CENP-W, which, in other eukaryotes, heterodimerizes with CENP-T via their respective histone fold domains (HFDs). The authors find that the C-terminal fragment of *Spodoptera frugiperda* forms a monomer in solution and is predicted, along with other insect CENP-T HFDs, to assume an unusual conformation whereby the $\alpha 3$ helix is oriented parallel to $\alpha 2$. The authors argue that this conformation, which brings the histone fold extension (HFE) closer to the main HFD stabilizes the monomer, effectively substituting for a dimerization partner.

The manuscript is clearly written and the data presented in a lucid manner. If the suggested model is correct, this would be the first known instance where a histone-fold has evolved away from obligate dimerization, making this an important discovery of broad interest to chromatin biologists.

I have two major comments that I think should be addressed/discussed/further investigated:

Major comments:

1. The authors show that a sfCNEP-T fragment containing the HFD and the histone fold extension (HFE) behaves as a monomer in solution. The authors implicitly assume that this is reflective of the full-length protein, but do not demonstrate this. The tail of histones can affect their dimerization behaviour (see e.g. recent work by Zhao et al 2024 PLoS Computational Biology 20(1):e1011721) and the authors should - if they do not have data on full-length proteins - appropriately caveat the conclusion that the full-length protein is also a monomer. The same reservations apply to downstream analysis using the C-terminal fragment, e.g. its ability to bind DNA. Whether, for examples, findings about the involvement of specific patches in DNA binding hold in the presence of the N-terminal tail is not a given.
2. The authors propose that insect CENP-T exists as a stable monomer because of a conformational change in $\alpha 3$ (+HFE) that stabilizes HFD. This inference is principally based on Alphafold modelling of the monomer. While consistent with their finding that CENP-T is a monomer, relying (critically) on Alphafold to infer a monomeric HFD structure might be considered problematic, not least because Alphafold will spit out a folded monomer for any histone (eukaryotic, archaeal) that I've tried, even though we know that the monomer assumes its structure only in the context of its dimerization partner. This is not to say that I don't believe the prediction here, but that corroborating data beyond Alphafold predictions are arguably indispensable. The only orthogonal data the authors present in this regard are data from cross-linking mass spectrometry (CLMS). They argue that the pre-eminence of crosslinks within a radius of 30 Angstrom supports their model. I'm not qualified to comment on whether these CLMS results really do support the model or whether the contact pattern is equally consistent with a standard histone fold and the brevity with which the data are presented here do not help me understand this. I suggest that somebody with the appropriate expertise should have a look at this.
3. I think it would be important to state more clearly in the text (or Figure legends) whether all the models generated were from full-length proteins and that the N-terminus was simply removed for visualization purposes. This matters because insect CENP-Ts are very large (>1000 amino acids) and the HFD/HFE represent only a small fraction (~100AA) of the protein. Line 108 indicates that the full-length protein was modelled for bmCENP-T, but it is not clear whether this is also the case for other

proteins discussed here.

Minor comments:

- The authors highlight that HFDs normally exist as dimers in solution. It might be worth highlighting explicitly that this is also true for HFDs outside a nucleosomal context, for example in the case where HFDs are embedded - and act as dimerization motifs - in larger proteins such as nuclear factor Y and son of sevenless (perhaps just following on from the sentence ending on line 34).
- It would be nice to pinpoint the amino acid changes (presumably in $\alpha 2$, L2, and/or $\alpha 3$) that allow $\alpha 3$ to swivel into a position parallel to $\alpha 2$. What prevents this from happening in other histones/enables this move in insect CENP-T? Identification of the specific residues involved, coupled with (in silico) mutagenesis and AlphaMissense-type evaluation of structural impact and linking to natural variation observed across insects, might further help to support their model.
- Line 24: saying "present in certain prokaryotic proteins" is a bit incongruous since you already mentioned archaea. Perhaps replace with "certain bacterial proteins"
- I found myself wondering how CENP-T binds DNA in the context of the kinetochore until I got to the Discussion and the pointer to Figure S3G. For the non-experts like me, it might be nice to move this explanation into the Introduction.
- Line 52: "typical of HFD proteins" - I don't know what this clause is meant to say as it seems to apply that heterodimerization with CENP-W is typical of HFD proteins.
- Line 180: insert "CENP-T" after "conventional"
- Given the availability of AlphaFold3, did the authors attempt to model CENP-T in complex with DNA? If not, why not? If so, do the results support the conclusions presented in Figure 3?
- I'm not entirely sure why the HFD domain is referred to as "cryptic". What's cryptic about it?

Typographical issues:

- In a few instances (e.g. line 39) the citation formatting has gone awry and citations are duplicated
- Line 181: insert "a" before "reduction"
- Line 231: should be "functional constraint"

It is my policy to sign my reviews.

Tobias Warnecke

Referee #2:

Many of proteins containing the histone fold domain (HFD) form a dimer or further tetramer using HFD like canonical histones. Authors in this paper is characterizing lepidopteran kinetochore protein CENP-T, a HFD protein and found that it exists as a monomer protein. Based on structure prediction and some experimental data, they propose that lepidopteran CENP-T shows structure reorganization in the HFD helix 3, which may not need a binding partner. This HFD reorganization was found in other insects, which do not have a CENP-T binding partner CENP-W. Interestingly, lepidopteran CENP-T still has DNA binding activity using a basic patch in this protein.

As lepidopteran does not have CENP-A, which is an epigenetic marker for centromere specification in many organisms, it is possible that CENP-T plays a role for connection of centromeric chromatin with the kinetochore complex in lepidopteran. Vertebrate CENP-T forms a dimer with CENP-W, and CENP-TW further form a CENP-TWSX tetramer. However, there appears to be no CENP-W in lepidopteran, and it is unclear how lepidopteran CENP-T binds to centromeric DNA. Here, authors demonstrated that lepidopteran CENP-T exists as a monomer and binds to DNA as a monomer form. This is potentially interesting observation and authors' explanation why CENP-T behaves as a monomer by structure viewpoints might be reasonable. However, this reviewer feels that authors need additional tests to say importance of CENP-T in lepidopteran. For example, they can test importance of DNA binding activity of CENP-T in vivo. Without in vivo data the paper would be immature.

1. Authors should propose biological significance of monomer form of lepidopteran CENP-T. Although CENP-T is an important kinetochore protein in lepidopteran, authors demonstrated that CENP-I plays an important role in their previous report. Authors mentioned possible importance of CENP-T-CENP-HIK interaction in this paper. They can predict binding surface of CENP-T to CENP-HIK and test its significance in vivo.
2. Related to comment#1, they should test importance of DNA binding domain in vivo. They can test whether mutant CENP-T can rescue mitotic defects after CENP-T RNAi.

Referee #3:

Sankaranarayanan et al., report here the identification of a monomeric histone fold domain-containing protein CENP-T from

lepidopteran. Previous work from the Drinnenberg lab showed that the kinetochore, a multi-subunit microtubule-binding protein assembly essential for accurately segregating the genetic information from a mother cell to the daughter cells, is distinctly different in *Bombyx mori* (an experimental model system belonging to Lepidoptera) and the species belonging to Lepidoptera lack CENP-A and CENP-C, key centromeric proteins otherwise form the foundation for the assembly of a functional kinetochore. This work demonstrates how evolutionarily distinct bmCENP-T whose HFE is repositioned within the three-dimensional structure to accommodate the loss of CENP-W, the HFD-containing protein required to stabilise CENP-T in other organisms containing CENP T/W/S/X complex, a protein complex acting as one of the supporting pillars of the kinetochore. This is an interesting finding, providing a novel evolutionary insight into how structural variations in an extremely well-conserved Histone Fold Domain (HFD) compensate for the loss of its binding partner.

Overall, this is a very simple but elegant study backed up with high-quality data supporting their claims.

Line 110: 'The C-terminus with the HFD and HFE were highly...' Rephrasing might help. I wonder if the authors mean, 'The C-terminal HFD and HFE were highly similar..'?

The results section starts with sfCENP-T and then suddenly switches to bmCENP-T. I suggest that the authors start the second section with sfCENP-T AF model before moving to bmCENP-T.

Although bmCENP-T is also highly likely to be a soluble monomer, demonstrating this by measuring the molecular weight using SEC-MALS (like Fig 1D) will strengthen the conclusion, particularly considering that, except for the first section (where the biochemical characterisation was on recombinant sfCENP-T), the rest of the manuscript revolves around bmCENP-T.

Figure S1b, why bmCENP-T runs faster than sfCENP-T in the SDS-PAGE? again, SEC-MALS of bmCENP-T would be helpful.

The validation of the AlphaFold model using crosslinking/MS could be strengthened by explicitly highlighting the percentage of crosslinking involving HFE residues that agree with the HFE-HFD interactions in the AlphaFold model mimicking CENP-T/W interaction.

Figure S2c, where the structural superposition of bmCENP-T and sfCENP-T is shown, it would be good also to include a figure similar to the one shown for sfCENP-T in Fig 2B, showing the hydrophobic core involving the HFE. In addition, including a multiple sequence alignment, like the one shown in Fig S3b, highlighting the amino acid residues forming the hydrophobic core with filled circles will be helpful to highlight the conserved nature of the hydrophobic interaction.

Figure S3C, The SEC profiles are confusing. It would help to include the peak elution volumes in the chromatogram itself. Why does mut1 elute earlier than WT (in between WT and mut3)? It is also important to maintain the same naming style in the figure as in the text. The authors should also include corresponding Coomassie-stained SDS PAGE gel showing the quality of the samples.

Referee #1:

Sankaranarayanan et al present a nicely focused exploration of altered properties of the CENP-T protein in insects, with a focus on Lepidopteran model species.

Along with CENP-A and CENP-C, insects have lost CENP-W, which, in other eukaryotes, heterodimerizes with CENP-T via their respective histone fold domains (HFDs). The authors find that the C-terminal fragment of *Spodoptera frugiperda* forms a monomer in solution and is predicted, along with other insect CENP-T HFDs, to assume an unusual conformation whereby the $\alpha 3$ helix is oriented parallel to $\alpha 2$. The authors argue that this conformation, which brings the histone fold extension (HFE) closer to the main HFD stabilizes the monomer, effectively substituting for a dimerization partner.

The manuscript is clearly written and the data presented in a lucid manner. If the suggested model is correct, this would be the first known instance where a histone-fold has evolved away from obligate dimerization, making this an important discovery of broad interest to chromatin biologists.

I have two major comments that I think should be addressed/discussed/further investigated:

We thank the reviewer for their positive evaluation. We appreciate their recognition of the potential significance of our findings and have addressed the major points in the revised manuscript, including the addition of new CLMS data that support the AlphaFold predicted structure.

Major comments:

1. The authors show that a sfCNEP-T fragment containing the HFD and the histone fold extension (HFE) behaves as a monomer in solution. The authors implicitly assume that this is reflective of the full-length protein, but do not demonstrate this. The tail of histones can affect their dimerization behaviour (see e.g. recent work by Zhao et al 2024 PLoS Computational Biology 20(1):e1011721) and the authors should - if they do not have data on full-length proteins - appropriately caveat the conclusion that the full-length protein is also a monomer. The same reservations apply to downstream analysis using the C-terminal fragment, e.g. its ability to bind DNA. Whether, for examples, findings about the involvement of specific patches in DNA binding hold in the presence of the N-terminal tail is not a given.

In the aforementioned study, the authors use several *in silico* methods to investigate the formation/stability of H2A/H2B heterodimers or H2A/H2A homodimers consisting of HFD only or containing the full-length proteins with their tails. The study is purely computational, and the results should therefore be viewed with caution as there is no experimental evidence for the existence of H2A/H2A homodimers or H2A/H2B heterodimers consisting only of HFD. Moreover, we have already demonstrated experimentally that the CENP-T HFD with the C-terminal tail (HFE) is monomer in solution. Nevertheless, it is theoretically possible that the full-length bmCENP-T would form dimers.

To address the reviewer's comment, we have generated a homodimeric structure of bmCENP-T with AlphaFold3 using the full-length sequence of the protein. The models generated for the homodimeric state of bmCENPT had very low confidence values (ipTM= 0.18 (preferably >0.8) and pTM= 0.22 (preferably >0.5)). The dimerization observed in these models did not occur via the histone folding domain as seen for typical HFD dimers. Instead, it was mediated by the solvent-exposed side of the HFE monomer (Figure EV2G). This was consistent across the top four models generated by AlphaFold. The same interface is involved in the interaction with the CENP-HIKM complex, its evolutionarily conserved interaction partner, for which AlphaFold3 predicts an interaction with much higher confidence (ipTM= 0.91 and pTM= 0.90) (Figure EV4D). Thus, dimerization by HFE appears to be physiologically unlikely. Despite these predictions, there is still a possibility that the full-length protein can form a dimer, yet unlikely in a manner reminiscent of canonical HFD interaction. We clearly state that the conclusions are for the C-terminal fragment with the HFD and extension, but not full-length CENP-T.

In the monomeric structure of full-length bmCENPT generated by AlphaFold, the presence of the disordered N-terminus had no effect on the folding/structure of the HFD (Figure EV2A). Based on our previous work (Cortes-Silva et al 2020) and other studies performed in vertebrates and fungi, we also know that the N-terminus of CENP-T is a conserved receptor of the DNA distal outer kinetochore complex and is present in an arrangement away from the HFD that binds DNA. Hence, we find it unlikely to influence the ability of these residues to bind DNA. Nevertheless, we tested the effect of these mutations on mitotic progression upon endogenous CENP-T depletion in *B. mori* cell lines (Figure EV4).

2. The authors propose that insect CENP-T exists as a stable monomer because of a conformational change in $\alpha 3(+HFE)$ that stabilizes HFD. This inference is principally based on AlphaFold modelling of the monomer. While consistent with their finding that CENP-T is a monomer, relying (critically) on AlphaFold to infer a monomeric HFD structure might be considered problematic, not least because AlphaFold will spit out a folded monomer for any histone (eukaryotic, archaeal) that I've tried, even though we know that the monomer assumes its structure only in the context of its dimerization partner. This is not to say that I don't believe the prediction here, but that corroborating data beyond AlphaFold predictions are arguably indispensable. The only orthogonal data the authors present in this regard are data from cross-linking mass spectrometry (CLMS). They argue that the pre-eminence of crosslinks within a radius of 30 Angstrom supports their model. I'm not qualified to comment on whether these CLMS results really do support the model or whether the contact pattern is equally consistent with a standard histone fold and the brevity with which the data are presented here do not help me understand this. I suggest that somebody with the appropriate expertise should have a look at this.

In the revised manuscript, we further validate the model by CLMS using a zero-length crosslinker EDC which provides sufficient resolution to validate the position of bmCENPT-HFE as proposed by the AlphaFold model rather than the conventional CENP-T structure. The observed crosslinks agree with the distance between the residues of HFE and HFD domains of the *B. mori* CENP-T as predicted by AlphaFold, but would not be anticipated in the context of canonical CENP-T. In the revised manuscript, Figures 2E and 2F are updated to show the EDC crosslinks detected in our assay. This has now also been described in the corresponding section of the manuscript (L135-142).

"We used the zero-length crosslinker 1-Ethyl-3-(3-dimethylaminopropyl)carbodiimide hydrochloride (EDC) to identify crosslinks between the HFD and HFE that are in proximity in the predicted structure of the altered HFD (10-13 Å, within the EDC crosslinking range). If the actual position of HFE is like that of a canonical fold, it would be beyond the crosslinking range of EDC, making it an ideal crosslinker to distinguish between these two conformations (Figure 2E-F, Table EV1). MS analyses of the crosslinked protein revealed that 6 of the 10 crosslinks involving a K/D/E residue in HFE were with the residues in $\alpha 2-3$ HFD. In the AlphaFold model, these residues pairs were found to be 9.2- 15.7 Å apart indicating that the protein adopts a further compacted structure in solution. This validates the proximity of the HFE and its role in stabilising the HFD predicted structure. Overall, these results provide a molecular basis for the monomeric nature of bmCENP-T HFD fragment"

3. I think it would be important to state more clearly in the text (or Figure legends) whether all the models generated were from full-length proteins and that the N-terminus was simply removed for visualization purposes. This matters because insect CENP-Ts are very large (>1000 amino acids) and the HFD/HFE represent only a small fraction (~100AA) of the protein. Line 108 indicates that the full-length protein was modelled for bmCENP-T, but it is not clear whether this is also the case for other proteins discussed here.

We thank the reviewer for bringing this up as it can indeed lead to confusions. All the models reported in this study were generated for full-length proteins. The disordered N' sequences were hidden for visualization purposes. This has been clarified in the concerned text and figure legends of the revised manuscript.

Minor comments:

- The authors highlight that HFDs normally exist as dimers in solution. It might be worth

highlighting explicitly that this is also true for HFDs outside a nucleosomal context, for example in the case where HFDs are embedded - and act as dimerization motifs - in larger proteins such as nuclear factor Y and son of sevenless (perhaps just following on from the sentence ending on line 34).

This has been corrected in the revised manuscript (L34-35)

“Besides histones, the HFD is also commonly detected in proteins involved in transcription, DNA replication and repair, chromatin remodelling, as well as at the kinetochore (Figure 1A), wherein they also retain the dimeric state apart from their role in DNA binding (Hartlepp et al, 2005; Kamada et al, 2001; Gangloff et al, 2001; He et al, 2017).”

- It would be nice to pinpoint the amino acid changes (presumably in $\alpha 2$, L2, and/or $\alpha 3$) that allow $\alpha 3$ to swivel into a position parallel to $\alpha 2$. What prevents this from happening in other histones/enables this move in insect CENP-T? Identification of the specific residues involved, coupled with (in silico) mutagenesis and AlphaMissense-type evaluation of structural impact and linking to natural variation observed across insects, might further help to support their model.

From the Shannon's entropy profile presented in the figure EV5B, it is evident that loop 2 and the sequences flanking it (C' terminus of Helix 2 and N' of Helix 3) are more divergent regions of the protein. Barring the conservation of few residues on the solvent exposed part of the domain, we do not see any specific amino acid signatures that could be correlated with the evolution of the monomeric HFD variant. While a proline is present as the last amino acid of loop2 in several species, it is not invariant (Figure EV2F). The altered structure is still detected in proteins having other amino acids in this position. Despite sequence diversity, we still see a conservation of hydrophobic residues in this region and across the extension, especially the regions facing the HFD. This suggests that the selection of sequences is primarily by their ability to stabilize the protein by forming a hydrophobic core. We have now included a brief discussion of the sequence evolution in the revised manuscript.

“For residues corresponding to regions folded differently in the monomeric fold, we observe a marked increase in sequence diversity hinting at an ongoing evolutionary process or lower constraints on the amino acid identity in this region (Figure EV5B). While we did not detect any sequence signature specific to the altered fold, we find the hydrophobicity of several amino acids in the HFE and HFD conserved (Figure EV2F).”

- Line 24: saying "present in certain prokaryotic proteins" is a bit incongruous since you already mentioned archaea. Perhaps replace with "certain bacterial proteins"

We have modified the introduction according to the suggestion.

- I found myself wondering how CENP-T binds DNA in the context of the kinetochore until I got to the Discussion and the pointer to Figure S3G. For the non-experts like me, it might be nice to move this explanation into the Introduction.

The part of the discussion referred in the above comment describes the DNA binding region in bmCENPT. A general description is included in the introduction (following L46) in the revised manuscript.

“CENP-T binds DNA directly via a C-terminal HFD in the context of a CENP-TWSX nucleosome-like complex, and super-coils linker DNA, thereby sharing structural and functional properties of histones (Nishino et al, 2012a; Takeuchi et al, 2014a; Hori et al, 2008a). At the CCAN, this complex partially wraps DNA and bends it through the interactions made by the positively charged residues in and upstream of the HFD- $\alpha 1$ (Yatskevich et al, 2022; Takeuchi et al, 2014b)”

- Line 52: "typical of HFD proteins" - I don't know what this clause is meant to say as it seems to apply that heterodimerization with CENP-W is typical of HFD proteins.

It was intended to point out the HFD's dependence on a partner for stabilization. The sentence has been rephrased to "Like other HFD proteins, the stability, localisation and function of CENPT is dependent on dimerization with a partner, CENPW." (L53-54 in the revised manuscript) for clarity.

“Like other HFD proteins, the stability, localisation and function of CENP-T is dependent on dimerization with a HFD partner, CENP-W (Schleiffer et al, 2012; Hori et al, 2008b).”

- Line 180: insert "CENP-T" after "conventional"

We incorporated this change.

- Given the availability of AlphaFold3, did the authors attempt to model CENP-T in complex with DNA? If not, why not? If so, do the results support the conclusions presented in Figure 3?

AlphaFold3 failed to model an interaction between the bmCENPT-C' and the 12bp DNA used in this study. We see similar outcomes after changing the inputs to full-length bmCENPT and human α -satellite DNA. The results are provided below. This is in contrast with our experimental evidence that strongly support bmCENPT to bind DNA *in vitro*. This discrepancy could be due to the limitations of AlphaFold3 in modelling protein-DNA complexes because the patch1 residues described in this study occupy similar positions in canonical CENP-T that is known to bind DNA.

bmCENPT C' + 12 bp DNA

bmCENPT^{FL} + α -satellite DNA

- I'm not entirely sure why the HFD domain is referred to as "cryptic". What's cryptic about it?

The term was used to highlight the deviation from the highly conserved HFD architecture. For clarity, we have used the terms altered or monomeric HFD in the revised manuscript.

Typographical issues:

- In a few instances (e.g. line 39) the citation formatting has gone awry and citations are duplicated

- Line 181: insert "a" before "reduction"

- Line 231: should be "functional constraint"

These mistakes have been corrected in the revised version of the manuscript.

Referee #2:

Many of proteins containing the histone fold domain (HFD) form a dimer or further tetramer using HFD like canonical histones. Authors in this paper is characterizing lepidopteran kinetochore protein CENP-T, a HFD protein and found that it exists as a monomer protein. Based on structure prediction and some experimental data, they propose that lepidopteran CENP-T shows structure reorganization in the HFD helix α_3 , which may not need a binding partner. This HFD reorganization was found in other insects, which do not have a CENP-T binding partner CENP-W. Interestingly, lepidopteran CENP-T still has DNA binding activity using a basic patch in this protein.

As lepidopteran does not have CENP-A, which is an epigenetic marker for centromere specification in many organisms, it is possible that CENP-T plays a role for connection of centromeric chromatin with the kinetochore complex in lepidopteran. Vertebrate CENP-T forms a dimer with CENP-W, and CENP-TW further form a CENP-TWSX tetramer. However, there appears to be no CENP-W in lepidopteran, and it is unclear how lepidopteran CENP-T binds to centromeric DNA. Here, authors demonstrated that lepidopteran CENP-T exists as a monomer and binds to DNA as a monomer form. This is potentially interesting observation and authors' explanation why CENP-T behaves as a monomer by structure viewpoints might be reasonable. However, this reviewer feels that authors need additional tests to say importance of CENP-T in lepidopteran. For example, they can test importance of DNA binding activity of CENP-T in vivo. Without in vivo data the paper would be immature.

We thank the reviewer for their valuable suggestions. In response, we have added new data using lepidopteran cell lines to directly test the biological significance of CENP-T's DNA-binding activity and its interaction with the HIK complex (Figure EV4, Lines 181-200 in the revised manuscript) .

1. Authors should propose biological significance of monomer form of lepidopteran CENP-T. Although CENP-T is an important kinetochore protein in lepidopteran, authors demonstrated that CENP-I plays an important role in their previous report. Authors mentioned possible importance of CENP-T-CENP-HIK interaction in this paper. They can predict binding surface of CENP-T to CENP-HIK and test its significance in vivo.

To follow up the reviewer's suggestion we used AlphaFold to predict the binding surface between the bmCENP-T and bmCENP-HIK. These analyses revealed amino acids that are present in the tail following the two-helical extension to be important for this interaction. As we do not have antibodies that allow us to directly visualize the localization of components of the bmCENP-HIK complex endogenously we turned to an alternative system to test whether ectopically expressed CENP-T can be recruited to a LacI-GFP-CENP-I fusion protein bound to a genomically integrated *LacO* array in Sf9 cells. Using this assay, we find that full-length CENP-T is robustly recruited to the *LacO* array, whereas a truncated version lacking the tail fails to localize, suggesting that the CENP-T tail is necessary for recruitment. Our data thus support that the interaction between CENP-T and the CENP-HIK complex is conserved (refer Figure EV4D-E, L190-200 in the revised manuscript).

“Besides DNA binding, the HFE of CENP-T is also essential for its kinetochore localization through its interaction with CENP-K of the CENP-HIKM complex (McKinley et al, 2015a; Pekgoz Altunkaya et al, 2016b; Nishino et al, 2012a; Basilico et al, 2014b). With the observed rearrangements in lepidopteran CENP-T, we predicted the interface of this interaction using AlphaFold multimer (Figure EV4D) and tested for its functional conservation as below. In a stable Sf9 cell line that contains a genomically integrated LacO array, we ectopically expressed LacI-GFP-sfCENP-I in tandem with either 3xFLAG tagged sfCENP-TWT or sfCENP-T Δ tail wherein the predicted sfCENP-K-interacting part of the protein was deleted (Figure EV4D). The ability of sfCENP-TWT to interact with sfCENP-HIKM was supported by its localization to the LacO foci bound by LacI-GFP-sfCENP-I (Figure EV4E and F). A significantly diminished localization of sfCENP-T Δ tail to the LacO foci suggested that the HFE of CENP-T still retained its role in mediating CENP-T and CENP-HIKM interaction (Figure EV4E and F).”

2. Related to comment#1, they should test importance of DNA binding domain in vivo. They can test whether mutant CENP-T can rescue mitotic defects after CENP-T RNAi.

To address the second comment, we tested the ability of ectopically expressed bmCENP-T^{WT} or CENP-T^{patch1} to rescue the defects caused by depletion of endogenous CENP-T. This assay is based on previous observations in chicken and human cell lines wherein kinetochore function was compromised upon loss of CENP-T-DNA interaction, causing mitotic defects (McKinley et al., 2015; Nishino et al., 2012). In the case of *B. mori*, we find the DNA binding dispensable for mitotic function of CENP-T in *B. mori* cell lines (refer Figure EV4 A-C, L 181-190 in the revised manuscript). This deviation and the possible explanations and limitations have been discussed in the revised manuscript (L251-260 in the revised manuscript).

“To understand the functional relevance of patch1 residues for mitotic progression in cells, we tested whether ectopically expressed bmCENP-TWT or bmCENP-Tpatch1 bearing the same substitutions tested in our electrophoretic mobility shift assays can rescue mitotic defects observed upon endogenous CENP-T depletion in a B. mori derived cell line BmN4-Sid1 (Kobayashi et al, 2012). As previously described (Cortes-Silva et al, 2020a), RNAi-mediated depletion of endogenous bmCENP-T resulted in mitotic arrest, seen in the form of congression and metaphase alignment defects (Figure EV4A-C). As expected, these defects were rescued by the expression of an RNAi-resistant version of bmCENP-TWT. Notably, we also find the RNAi-resistant version of bmCENP-Tpatch1 to have a similar effect, suggesting that the DNA binding ability of bmCENP-T is not essential for accurate mitotic progression at least in a B. mori cell line.”

Referee #3:

Sankaranarayanan et al., report here the identification of a monomeric histone fold domain-containing protein CENP-T from lepidopteran. Previous work from the Drinnenberg lab showed that the kinetochore, a multi-subunit microtubule-binding protein assembly essential for accurately segregating the genetic information from a mother cell to the daughter cells, is distinctly different in *Bombyx mori* (an experimental model system belonging to Lepidoptera) and the species belonging to Lepidoptera lack CENP-A and CENP-C, key centromeric proteins otherwise form the foundation for the assembly of a functional kinetochore. This work demonstrates how evolutionarily distinct bmCENP-T whose HFE is repositioned within the three-dimensional structure to accommodate the loss of CENP-W, the HFD-containing protein required to stabilise CENP-T in other organisms containing CENP T/W/S/X complex, a protein complex acting as one of the supporting pillars of the kinetochore. This is an interesting finding, providing a novel evolutionary insight into how structural variations in an extremely well-conserved Histone Fold Domain (HFD) compensate for the loss of its binding partner. Overall, this is a very simple but elegant study backed up with high-quality data supporting their claims.

We thank the reviewer for their positive assessment and appreciation of our study.

Line 110: 'The C-terminus with the HFD and HFE were highly...' Rephrasing might help. I wonder if the authors mean, 'The C-terminal HFD and HFE were highly similar..!'

We have now rephrased the sentence according to the suggestion.

“To understand the molecular basis of this stability, we used AlphaFold to model the structure of full-length CENP-T from B. mori, the most established lepidopteran model organism in which CENP-T was functionally characterized (Figure 2A). While the N-terminus was largely disordered, the C-terminal HFD and HFE adopted a similar structure across different AlphaFold models and were generated with high confidence (pLDDT>90) (Figure EV2A-C).”

The results section starts with sfCENP-T and then suddenly switches to bmCENP-T. I suggest that the authors start the second section with sfCENP-T AF model before moving to bmCENP-T.

We considered the reviewer's suggestion. In the revised manuscript, we now added SEC-MALS data for the *B. mori* CENP-T C-terminus to the first section (Figure EV1C, L106), complementing the already included data for *S. frugiperda*. As a result, *B. mori* is now introduced earlier in the text, and we felt it was more logical to retain the current structure of the second section.

“The CENP-T-HFD from second lepidopteran species Bombyx mori was also found to be a soluble monomeric protein, further strengthening our observations (Figure EV1B-C). These findings corroborate that the HFD in lepidopteran CENP-T folds independent of an interacting partner and is a monomer in solution.”

Although bmCENP-T is also highly likely to be a soluble monomer, demonstrating this by measuring the molecular weight using SEC-MALS (like Fig 1D) will strengthen the conclusion, particularly considering that, except for the first section (where the biochemical characterisation was on recombinant sfCENP-T), the rest of the manuscript revolves around bmCENP-T.

We agree with the reviewer’s suggestion. We experimentally verified the monomeric nature of bmCENP-T-HFD by SEC-MALS analyses. This has been included in the results section (refer to Figure EV1C in the revised manuscript).

Figure S1b, why bmCENP-T runs faster than sfCENP-T in the SDS-PAGE? again, SEC-MALS of bmCENP-T would be helpful.

The fragments containing the HFD cloned for sfCENP-T and bmCENP-T are different (bmCENP-T- 16.2 kDa, sfCENP-T- 21.5 kDa), hence evident in their rates of migration. We now have experimental evidence that both these fragments are monomer in solution. This has been described in the response to the previous comment.

The validation of the AlphaFold model using crosslinking/MS could be strengthened by explicitly highlighting the percentage of crosslinking involving HFE residues that agree with the HFE-HFD interactions in the AlphaFold model mimicking CENP-T/W interaction.

We thank the reviewer for suggesting this metric for testing the AlphaFold model. Of the 60% of all the crosslinks that emerge from K/D/E in the HFE (10 crosslinks in total) map to residues in the α 2-3 of HFD. Furthermore, these residue pairs are ~9.5-15 Å apart in the model. This suggests that the protein adopts a compact structure in solution and supports the proximity between the HFE and the HFD as predicted by AlphaFold. The above inference has been included in the revised manuscript (L135-142).

“We used the zero-length crosslinker 1-Ethyl-3-(3-dimethylaminopropyl)carbodiimide hydrochloride (EDC) to identify crosslinks between the HFD and HFE that are in proximity in the predicted structure of the altered HFD (10-13 Å, within the EDC crosslinking range). If the actual position of HFE is like that of a canonical fold, it would be beyond the crosslinking range of EDC, making it an ideal crosslinker to distinguish between these two conformations (Figure 2E-F, Table EV1). MS analyses of the crosslinked protein revealed that 6 of the 10 crosslinks involving a K/D/E residue in HFE were with the residues in α 2-3 HFD. In the AlphaFold model, these residues pairs were found to be 9.2- 15.7 Å apart indicating that the protein adopts a further compacted structure in solution. This validates the proximity of the HFE and its role in stabilizing the HFD predicted structure. Overall, these results provide a molecular basis for the monomeric nature of bmCENP-T HFD fragment”.

Figure S2c, where the structural superposition of bmCENP-T and sfCENP-T is shown, it would be good also to include a figure similar to the one shown for sfCENP-T in Fig 2B, showing the hydrophobic core involving the HFE. In addition, including a multiple sequence alignment, like the one shown in Fig S3b, highlighting the amino acid residues forming the hydrophobic core with filled circles will be helpful to highlight the conserved nature of the hydrophobic interaction.

The hydrophobic core formed by sfCENP-T is presented in Figure EV2E, and the conservation of hydrophobic residues in a subset of lepidopterans is presented in figure EV2F in the revised manuscript.

Figure S3C, The SEC profiles are confusing. It would help to include the peak elution volumes in the chromatogram itself. Why does mut1 elute earlier than WT (in between WT and mut3)? It is also important to maintain the same naming style in the figure as in the text. The authors should also include corresponding Coomassie-stained SDS PAGE gel showing the quality of the samples.

The elution profile from size exclusion chromatography was used as a measure to compare the fold of the different mutants relatively between themselves and compared to the wild type. While mutations in patch 2 showed less impact on the fold, mutating all

residues in patch 1 resulted in slight delay in the elution volume, probably by perturbing the N-terminal loop and Helix conformation/fold. Mutations in patch 3 had more drastic effect on the fold based on the GF with an approximative estimated molecular weight of 29 kDa based on the calibration curve, suggesting more important unfolding of the protein or unnatural oligomerized status. SDS-PAGE gels for size exclusion chromatography have been added for all mutants to show the protein quality (Figure EV3D).

Dear Dr. Drinnenberg

Thank you for the submission of your revised manuscript to our offices. We have now received the enclosed reports from the referees that were asked to assess it. EMBOR-2024-60787V2 still has suggestions by Referee #2 that I would like you to incorporate and address before we can accept your manuscript for publication. Please make sure to include a response letter in your revision providing your responses to the few remaining issues raised by Referee #2.

As you will see there are also important comments and concerns raised by our editorial assistants and data integrity team that you would need to address. While detailed responses to this do not need to appear in your response letter, it is crucial to fully address all these concerns. Should something be unclear or require further clarification, please do not hesitate to contact us.

I look forward to seeing a new revised version of your manuscript as soon as possible. Please use this link to submit your revision: <https://embor.msubmit.net/cgi-bin/main.plex>

I have noticed that the manuscript does not contain a materials and methods section. Basic materials and methods essential to the understanding of the experiments must be described in the main body of the manuscript and may not be presented as supplementary information. Please refer to my previous letter for more details in our length constrains.

Data of gene expression experiments described in submitted manuscripts should be deposited in a MIAME-compliant format with one of the public databases. We would therefore ask you to submit your microarray data to the ArrayExpress database maintained by the European Bioinformatics Institute for example. ArrayExpress allows authors to submit their data to a confidential section of the database, where they can be put on hold until the time of publication of the corresponding manuscript. Please see <https://www.ebi.ac.uk/arrayexpress/Submissions/> or contact the support team at arrayexpress@ebi.ac.uk for further information.

Yehu Moran
Academic Editor
EMBO Reports

Specific notes by editorial assistants:

MANUSCRIPT FORMAT: NOT OK - Methods not included in the main text, but in another Word with Reagents; Methods section needs to be included in the manuscript, after Discussion section

Keywords: missing. Please complete.

Data availability statement: Now added to the file with the Methods, needs to be in the main manuscript file, as well as the Methods section; direct URL needs to be provided for the PXD063732 deposition.

Conflict of interest/DCIS: missing

AUTHORS: The corresponding author needs to be clearly labeled on the title page, that is, we need the email address of the corresponding author included on the title page.

CHECKLIST: included, but missing a response in cell D114. Please correct.

FUNDING INFO: the info is matching, however, the funders and grants in the Comments box need to be removed and each should be uploaded in the system as a separate entry via More Funders option.

FIGURE CALLOUTS: Table S2 called out but missing - could this be the table with primers in the Related ms file? In any case the callout is not correct; "supplemental information" needs to be updated as we are no longer using this nomenclature.

DATASET EV LEGENDS: there are 2 EV tables, but they look like datasets - so the nomenclature needs to be updated in all places (source file names, titles in submission systems, callouts in the manuscript) to Dataset EV1 and Dataset EV2.

SYNOPSIS IMAGE: missing. Please provide

SYNOPSIS TEXT: missing. Please provide.

R&T TABLE: in the Related manuscript file with the Methods - we need a separate file uploaded (the template is in the GTA).

SOURCE DATA: SD is uploaded with completed checklist; SD needs to be uploaded as one folder per figure; each figure folder should have the the files with panels; Source data files need to be submitted as zipped folders, one .zip folder for each figure. Inside each folder, the files should be organized in subfolders, one subfolder for each panel. The assistant adds that SD files now re-organized and 1B has been provided as a separate file with some other SD panels; each panel of this file needs to be provided in its respective folder 9.5.25 BP

DATA CHECK: FAIL - Please note that the data availability statement is not provided in the manuscript.

==>DAS now added to the file with the Methods, needs to be in the manuscript file, as well as the Methods section; direct URL needs to be provided for the PXD063732 deposition.

Figure Legends - Comments

- Please define the annotated p values ****/****/**/* as well as provide the exact p-values for the same in the legend of figure EV4 F as appropriate.

- Please note that the box plots need to be defined in terms of minima, maxima, centre, bounds of box and whiskers, and percentile in the legend of figure EV4 F

Report regarding Data Integrity:

A potential reuse of blot panels between Figure 1C and Figure EV1A, which is not currently mentioned in the figure legend. This requires clarification from the authors.

Please review the issue and provide justification for reuse. If it is warranted, the duplication can remain, but it must be clearly indicated in the figure legends and the reason should be mentioned.

REFEREE REPORTS:

Referee #1:

The authors have addressed my comments in a satisfactory manner.

Referee #2:

The authors revised the manuscript on insect monomeric CENP-T based on the reviewers' comments. I appreciate their efforts to address the reviewers' concerns. However, I still have concerns about the data from the additional experiments. They should address my points with additional experiments.

In my previous comments, I suggested additional experiments on the functions of CENP-T *in vivo*. The authors performed two experiments. One experiment tried to identify the CENP-HIK binding region in CENP-T. The authors demonstrated that the CENP-T tail-less mutant did not recruit CENP-I based on LacO-LacI experiments in insect cells. The second experiment attempted to show whether the CENP-T DNA-binding mutant could rescue the CENP-T knockdown (KD) phenotype. The authors concluded that DNA binding may not be essential for CENP-T function.

1. For the LacO-LacI experiments, the bottom image with the CENP-T tail-less in Figure EV4E is not ideal. It should show a single GFP signal in one nucleus. However, there are three round signals in the image. They seem to be nonspecific aggregates. They should use a clearer image like CENP-T wild-type sample (upper in Figure EV4E).
2. The DNA binding of CENP-T may be dispensable for CENP-T functions, suggesting that the centromere localization of CENP-T depends on other factors. Since the authors demonstrated that the CENP-T tail is responsible for the CENP-HIK interaction, they should test whether the CENP-T tail-less mutant can rescue the CENP-T KD phenotype. This is an important experiment for discussing CENP-T function *in vivo*.

Referee #3:

The authors have revised their original manuscript with a substantial amount of additional data, further strengthening their conclusions and improving the overall quality of the manuscript. Hence, I am happy to recommend the revised manuscript for publication.

Response to reviewers comments

We thank all the reviewers for their constructive criticism and suggestions that helped us improve the quality of this manuscript. Below are the point-wise response to comments from reviewer 2.

Referee #2:

The authors revised the manuscript on insect monomeric CENP-T based on the reviewers' comments. I appreciate their efforts to address the reviewers' concerns. However, I still have concerns about the data from the additional experiments. They should address my points with additional experiments.

In my previous comments, I suggested additional experiments on the functions of CENP-T in vivo. The authors performed two experiments. One experiment tried to identify the CENP-HIK binding region in CENP-T. The authors demonstrated that the CENP-T tail-less mutant did not recruit CENP-I based on LacO-LacI experiments in insect cells. The second experiment attempted to show whether the CENP-T DNA-binding mutant could rescue the CENP-T knockdown (KD) phenotype. The authors concluded that DNA binding may not be essential for CENP-T function.

1. For the LacO-LacI experiments, the bottom image with the CENP-T tail-less in Figure EV4E is not ideal. It should show a single GFP signal in one nucleus. However, there are three round signals in the image. They seem to be nonspecific aggregates. They should use a clearer image like CENP-T wild-type sample (upper in Figure EV4E).

We have replaced the image panels for EV4E to show cells where there is a single LacO-GFP signal to avoid ambiguity. However, we would like to point out that these cell lines are polyclonal and may have more than one copy of the LacO array integrated in them, which are often detected as multiple foci.

2. The DNA binding of CENP-T may be dispensable for CENP-T functions, suggesting that the centromere localization of CENP-T depends on other factors. Since the authors demonstrated that the CENP-T tail is responsible for the CENP-HIK interaction, they should test whether the CENP-T tail-less mutant can rescue the CENP-T KD phenotype. This is an important experiment for discussing CENP-T function in vivo.

As suggested by the reviewer, we generated BmN4 cell lines that expressed a tail-less version of CENPT ($bmCENP-T^{\Delta tail}$) and tested their ability to complement the depletion of endogenous CENP-T. Concurrent with the observations in Sf9 cells, the tail-less version of $bmCENP-T$ failed to localize to the kinetochores, thereby mimicking CENP-T depletion phenotype. The results from this experiment are included in Figure EV4A-B. The corresponding section can be found in L204-211 and L272-274 of the revised manuscript.

Response to specific notes by editorial assistants:

- MANUSCRIPT FORMAT: NOT OK - Methods not included in the main text, but in another Word with Reagents; Methods section needs to be included in the manuscript, after Discussion section

Methods included in the main manuscript file in the revised version

Keywords: missing. Please complete.: *Included in the revised version*

Data availability statement: Now added to the file with the Methods, needs to be in the main manuscript file, as well as the Methods section; direct URL needs to be provided for the PXD063732 deposition.

Included in the revised version. The deposition is not accessible until the manuscript is published online. Hence we are unable to provide a URL. However, we have included a link and credentials with which reviewers can access the submission.

Conflict of interest/DCIS: missing: *Included in the revised version*

AUTHORS: The corresponding author needs to be clearly labeled on the title page, that is, we need the email address of the corresponding author included on the title page.

Corrected

CHECKLIST: included, but missing a response in cell D114. Please correct.

Corrected

FUNDING INFO: the info is matching, however, the funders and grants in the Comments box need to be removed and each should be uploaded in the system as a separate entry via More Funders option.

Corrected

FIGURE CALLOUTS: Table S2 called out but missing - could this be the table with primers in the Related ms file? In any case the callout is not correct; "supplemental information" needs to be updated as we are no longer using this nomenclature.

All the figure/table citations have been verified and renamed to EV wherever necessary.

DATASET EV LEGENDS: there are 2 EV tables, but they look like datasets - so the nomenclature needs to be updated in all places (source file names, titles in submission systems, callouts in the manuscript) to Dataset EV1 and Dataset EV2.

Corrected

SYNOPSIS IMAGE: missing. Please provide

Included in the revised submission

SYNOPSIS TEXT: missing. Please provide.

Included in the revised submission

R&T TABLE: in the Related manuscript file with the Methods - we need a separate file uploaded (the template is in the GTA).

Included in the revised submission

SOURCE DATA: SD is uploaded with completed checklist; SD needs to be uploaded as one folder per figure; each figure folder should have the the files with panels; Source data files need to be submitted as zipped folders, one .zip folder for each figure. Inside each folder, the files should be organized in subfolders, one subfolder for each panel. The assistant adds that SD files now re-organized and 1B has been provided as a separate file with some other SD panels; each panel of this file needs to be provided in its respective folder 9.5.25 BP

Uploaded figure wise in the revised submission

DATA CHECK: FAIL - Please note that the data availability statement is not provided in the manuscript.

Included in the revised version

==>DAS now added to the file with the Methods, needs to be in the manuscript file, as well as the Methods section; direct URL needs to be provided for the PXD063732 deposition.

Figure Legends - Comments

- Please define the annotated p values ****/***/**/* as well as provide the exact p-values for the same in the legend of figure EV4 F as appropriate.
- Please note that the box plots need to be defined in terms of minima, maxima, centre, bounds of box and whiskers, and percentile in the legend of figure EV4 F

The necessary information is Included in the legend for EV4F in the revised version

Report regarding Data Integrity:

A potential reuse of blot panels between Figure 1C and Figure EV1A, which is not currently mentioned in the figure legend. This requires clarification from the authors. Please review the issue and provide justification for reuse. If it is warranted, the duplication can remain, but it must be clearly indicated in the figure legends and the reason should be mentioned.

Addressed- this has been clearly stated in both the figure legends.

Dr. Ines Drinnenberg
Institut Curie
UMR3664
Paris
France

Dear Dr. Drinnenberg,

I am pleased to inform you that your manuscript has been accepted for publication in EMBO Reports. Your manuscript will be processed for publication by EMBO Press. It will be copy edited and you will receive page proofs prior to publication. Please note that you will be contacted by Springer Nature Author Services to complete licensing and payment information.

Yours sincerely,

Yehu Moran
Academic Editor
EMBO Reports

Referee #1:

The authors had previously answered all my comments and I have no further comments to make.

Referee #2:

This is an additional revision of "Insect CENP-T" by Drinnenberg and her colleagues. They addressed my previous comments clearly. The paper is now acceptable.

There is one minor point.

There are many duplicates in the reference list (e.g., Hori et al., 2008a and b; McKinley et al., 2015a and b; Nishino et al., 2012a and b; Pekgoz Altunkaya et al., 2016a and b; Schleiffer et al., 2012a and b; Takeuchi et al., 2014a and b). Please carefully check the reference list and correct these errors.

Referee #3:

In my opinion, the authors have satisfactorily addressed the concerns raised by the reviewers. I recommend the manuscript for publication.
